# Absence of photophysiological response to iron addition in autumn phytoplankton in the Antarctic Sea-Ice Zone

Asmita Singh[1,2], Susanne Fietz[1], Sandy J. Thomalla[2], Nicolas Sanchez[3], Murat V. Ardelan[3], Sébastien Moreau[4,5], Hanna M. Kauko[4], Agneta Fransson[4], Melissa Chierici[6], Saumik Samanta[1], Thato N. Mtshali[7], Alakendra N. Roychoudhury[1] and Thomas J. Ryan-Keogh[2]

[1] Department of Earth Sciences, University of Stellenbosch, Stellenbosch, South Africa
[2] Southern Ocean Carbon-Climate Observatory, CSIR, Cape Town, South Africa
[3] Department of Chemistry, Norwegian University of Science and Technology (NTNU), Trondheim, Norway
[4] Norwegian Polar Institute (NPI), Tromsø, Norway
[5] Centre for Ice, Cryosphere, Carbon and Climate, Department of Geosciences, UiT, The Arctic University of Norway, Tromsø, Norway
[6] Institute of Marine Research, Fram Centre, Tromsø, Norway
[7] Oceans and Coast, Department of Environment, Forestry and Fisheries, Cape Town, South Africa

*Correspondence to*: Thomas J. Ryan-Keogh (tryankeogh@csir.co.za)

**Abstract.** The High Nutrient-Low Chlorophyll condition of the Southern Ocean is generally thought to be caused by the low bioavailability of micronutrients, particularly iron, which plays an integral role in phytoplankton photosynthesis. Nevertheless, the Southern Ocean experiences seasonal blooms that generally initiate in austral spring, peak in summer and extend into autumn. This seasonal increase in primary productivity is typically linked to the seasonal characteristics of nutrient and light supply. To better understand the potential limitations on productivity in the Antarctic Sea-Ice Zone (SIZ), the photophysiological response of phytoplankton to iron addition (2.0 nM $FeCl_3$) was investigated during autumn along the Antarctic coast off Dronning Maud Land. Five short-term (24 hr) incubation experiments were conducted around Astrid Ridge (68°S) and along a 6°E transect, where an autumn bloom was identified in the region of the western SIZ. Surface iron concentrations ranged from 0.27 to 1.39 nM around Astrid Ridge, and 0.56 to 0.63 nM along the 6°E transect. Contrary to expectation, the photophysiological response of phytoplankton to iron addition, measured through the photosynthetic efficiency and the absorption cross-section for photosystem II, showed no significant responses. It is thus proposed that since the autumn phytoplankton in the SIZ exhibited a lack of an iron limitation at the time of sampling, the ambient iron concentrations may have been sufficient to fulfil the cellular requirements. This provides new insights into extended iron replete post-bloom conditions in the typically assumed iron deficient High Nutrient-Low Chlorophyll Southern Ocean.

## 1 Introduction

The Southern Ocean plays an important role in the global drawdown of atmospheric carbon dioxide ($CO_2$) (Khatiwala et al., 2009; Takahashi et al., 2002; 2009), which is partially driven by the biological carbon pump through phytoplankton photosynthetic carbon uptake and export. Seasonal changes in the physical and chemical environment of the Southern Ocean are expected to modify the physiological (Deppeler and Davidson 2017; Moore et al., 2013) and metabolic functions of phytoplankton and consequently the efficiency of the biological carbon pump (Boyd et al., 2007; 2010b). The primary factors that limit carbon fixation during phytoplankton photosynthesis in the Southern Ocean are the availability of light (Kirk, 1994; de Baar et al., 2005; Trimborn et al., 2019) and several essential trace metals (Sunda, 1989; Lindsey and Scott, 2010; Wu et al., 2019; Browning et al., 2021; Hawco et al., 2022), particularly iron, which is a crucial co-factor for the functioning of photosynthetic proteins (Raven, 1990; Raven et al., 1999; Strzepek and Harrison, 2004). In addition, iron is needed for nitrate reductase, which is responsible for the reduction of nitrate to nitrite (Sunda, 1989; Milligan and Harrison, 2000; de Baar et al., 2005; Bazzani et al., 2023), and is also required for the synthesis of chlorophyll and the quenching of reactive oxygen species (Sunda and Huntsman, 1995; Diaz and Plummer, 2018). However, nitrate assimilation has a high iron (Milligan and Harrison, 2000; de Baar et al., 2005) and light (Lucas et al., 2007; Moore et al., 2007a; 2007b) demand, which drives the High Nutrient-Low Chlorophyll (HNLC) conditions characteristic of the Southern Ocean (Price et al., 1994; Milligan and Harrison, 2000; Lucas et al., 2007; Cochlan, 2008; Moore et al., 2013). Thus, independent of adequate amounts of macronutrient concentrations in surface waters, any limitation on the bioavailability of iron will potentially decrease the efficiency of these processes (Martin and Fitzwater, 1988; Moore et al., 2001; Lis et al., 2015; Yoon et al., 2018), affecting nutrient drawdown, photosynthesis, primary productivity, biomass accumulation, and community composition of surface phytoplankton in the Southern Ocean (de Baar et al., 1990; Geider and La Roche, 1994; Martin et al., 1991; Martin and Fitzwater, 1988; Biggs et al., 2022). Furthermore, any light limitation will exacerbate iron limitation due to the increase in iron demand under low light conditions (Strzepek et al., 2012; 2019; Boyd and Abraham, 2001), thus driving the frequent occurrence of iron-light co-limitation conditions in the Southern Ocean (Moore et al., 2013; Tagliabue et al., 2014; Ryan-Keogh et al., 2017a; Trimborn et al., 2019).

Although the Southern Ocean is typically considered an "iron-limited" region, iron availability or limitation is not uniform, and instead varies spatially and temporally. For instance, iron limitation is commonly associated with the pelagic waters of the Southern Ocean (Mitchell et al., 1991; Yoon et al., 2018), where summer dissolved iron (dFe) concentrations in surface waters are typically <0.5 nM (Sedwick et al., 1999;

Coale et al., 1999; Vink and Measures, 2001; Klunder et al., 2011); however, there are a number of regional exceptions. These include regions with an external iron source such as sea-ice and iceberg meltwaters (Lannuzel et al., 2008; Boyd and Ellwood, 2010; Smith et al., 2010; Boyd et al., 2012), hydrothermal vents (Klunder et al., 2011; Tagliabue et al., 2017; Ardyna et al., 2019), atmospheric dust (Martin and Fitzwater, 1988; Mahowald et al., 2005), continental margins input (Sedwick et al., 2008; Bowie et al., 2009) and island wake inputs (Pollard et al., 2007; Blain et al., 2008). Internal processes such as remineralization (Tagliabue et al., 2017), resupply through deep winter mixing (Tagliabue et al., 2014), cross-frontal mixing (Lutjeharms et al., 1985; Moore and Abbott, 2002) and storm-driven entrainment (Nicholson et al., 2019) can also provide iron to surface waters in support of phytoplankton production. Most of these sources vary seasonally; for example, in winter, iron is not generally considered limiting as deep winter mixing entrains a seasonal resupply of iron (Tagliabue et al., 2014; Mtshali et al., 2019). Instead, due to the deep seasonal mixed layers, ice cover and low sun angles, the availability of photosynthetically active radiation (PAR) can be suboptimal and considered the dominant factor limiting phytoplankton production in winter. In spring, phytoplankton blooms are initiated when there is sufficient light, driven by a shoaling of the mixed layer (Moore and Abbott, 2002; Thomalla et al., 2011) as well as retreating sea-ice (Taylor et al., 2013) to support phytoplankton growth under nutrient replete conditions (Swart et al., 2015; de Baar et al., 1990; Hauck et al., 2015; Martin et al., 1990). Blooms typically subside when nutrients such as iron are depleted in late summer or early autumn (Tagliabue et al., 2014; Soppa et al., 2016; Hiscock et al., 2008). Grazing (Lancelot et al., 1993; Moreau et al., 2020; Kauko et al., 2021), bacteria and viruses (Biggs et al., 2021) may also accelerate the blooms' demise. Iron supply mechanisms during a bloom, such as advection from continental margins (Sedwick et al., 2008; Bowie et al., 2009), remineralization (Tagliabue et al., 2017) and storm-driven entrainment (Swart et al., 2015; Nicholson et al., 2019) may sustain phytoplankton growth for an extended duration. However, it is not clear how applicable these resupply processes are to the Southern Ocean as a whole, and where and when each of these dominate.

In general, experiments that investigate the degree of iron limitation by testing the impact of iron addition on metabolic functions of phytoplankton have largely focussed on summer conditions in the open Southern Ocean. There is thus minimal information on the impact of iron addition in the Sea-Ice Zone (SIZ) in autumn, when iron concentrations are expected to be low (Tagliabue et al., 2014; Lannuzel et al., 2016). One exception was a study by Van Oijen et al. (2004), where a single iron-light perturbation experiment examined carbon uptake in the marginal ice zone in autumn, but no conclusions were made on the driving factors of enhanced uptake. To address this knowledge gap, we undertook a number of iron addition experiments using active chlorophyll-a (Chl-a) fluorescence in the SIZ off Dronning Maud Land (DML) in autumn (March).

Active Chl-a fluorescence is a key indicator of the photophysiological state of phytoplankton (Hughes et al., 2018; Brown et al., 2019; Schuback et al., 2021) and provides a powerful tool for evaluating the photophysiological response of phytoplankton to iron addition. This is done by measuring the photosynthetic efficiency, $F_v/F_m$, and the absorption cross-section of photosystem II, $\sigma_{PSII}$ (Geider, 1993; Geider and La Roche, 1994; Kolber et al., 1988; 1994; Hughes et al., 2018). Any photophysiological response measured through active Chl-a fluorescence can, however, be due to both changes in cellular structure, i.e., a response seen on short timescales (milliseconds to femtoseconds), and changes in community composition, i.e., a response seen on longer time scales (usually >24 hrs). Since different phytoplankton groups tend to have different photophysiological signatures (Suggett et al., 2009), any measured response in photophysiology over longer time periods (>24 hrs) is difficult to interpret as it reflects both the cellular and community adjustments. This makes it difficult to resolve the physiological response of phytoplankton to iron addition in manipulation incubation experiments from community composition adjustments (Suggett et al., 2009).

Many iron addition incubation experiments previously conducted in the Southern Ocean (de Baar et al., 1990; Hinz et al., 2012; Ryan-Keogh et al., 2018; Viljoen et al., 2018; among others) were run for long time periods (>96 hrs), and showed evidence of substantial changes in community composition, which are likely to influence the photophysiological signal and consequently the interpretation of iron limitation (Ryan-Keogh et al., 2013; Suggett et al., 2009). In this paper, we opted instead for short-term (24 hr) incubation experiments to isolate changes in photophysiology, i.e., $F_v/F_m$ and $\sigma_{PSII}$. This is in line with a study by Ryan-Keogh (2014), which tested whether 24 hrs was sufficient to allow a measurable photophysiological response in Southern Ocean phytoplankton, where low temperatures may control uptake kinetics. Ryan-Keogh (2014) compared the photophysiology between incubations running for 24 and 48 hrs in summer and found that the samples were iron-limited (i.e., the differences between unamended control and iron addition incubations were significant). However, no significant differences were observed in photophysiology following iron addition when comparing the incubations of 24 vs 48 hrs, supporting the robustness of a representative response in photophysiology within 24 hrs. During this timeframe, the community composition is not expected to change, nor would we expect to see any adjustments in biomass or nutrient drawdown (Browning et al., 2014a; Ryan-Keogh et al., 2013; 2017a). As such, this study reported here, provides a unique investigation of the short-term photophysiological response of phytoplankton to iron addition in the SIZ in autumn, a season where iron limitation may be expected and a season and region that is under sampled. The experiments test the hypothesis that phytoplankton in the SIZ off DML experience iron limitation during post-bloom conditions in autumn.

## 2 Materials and methods

The focus of this study is on five short-term (24 hr) incubation experiments performed in March during the Southern Ocean Ecosystem Cruise (cruise number DML2019702) between 28 February and 10 April in 2019, on-board the Norwegian RV *Kronprins Haakon* in the SIZ of the Kong Håkon VII Hav off the Dronning Maud Land coast, as well as the region surrounding the Astrid Ridge (Fig. 1). Ancillary data (i.e., Chl-a concentrations, macronutrient concentrations and dFe concentrations) from surface water samples provide information on the regional conditions surrounding the five incubation experiments at the time of the cruise.

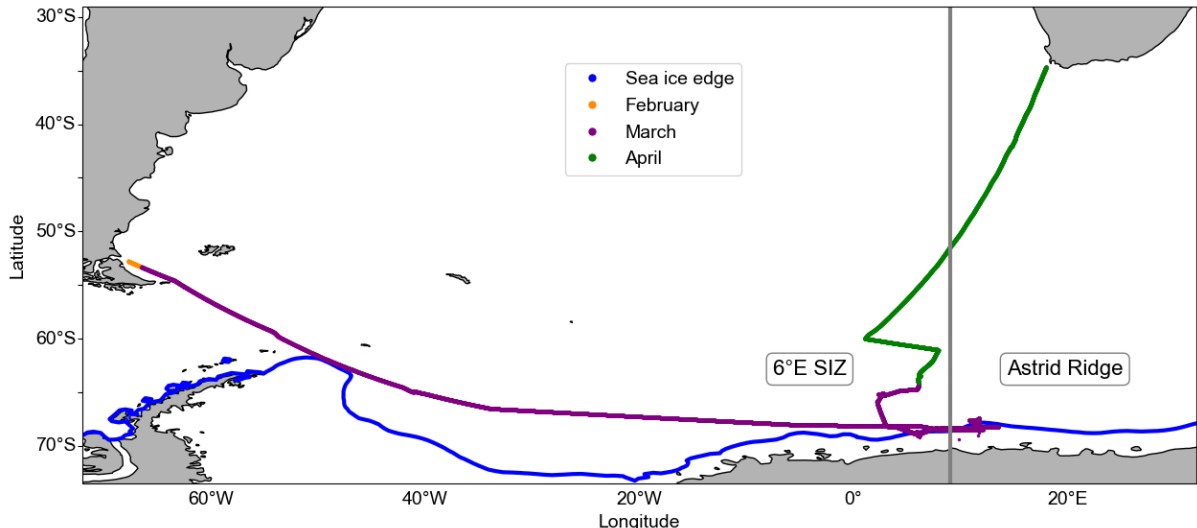

**Figure 1. Map of the general study region depicting the cruise track of the DML2019702 cruise that began in Punta Arenas, Chile, on 28 February 2019, traversed the Atlantic Southern Ocean and the Dronning Maud Land Sea ice edge in March, ending in Cape Town, South Africa on 10 April 2019. The 6°E SIZ and Astrid Ridge regions are indicated as well as the average sea ice edge (concentration at 15%) for March 2019 (Brodzik and Stewart, 2016).**

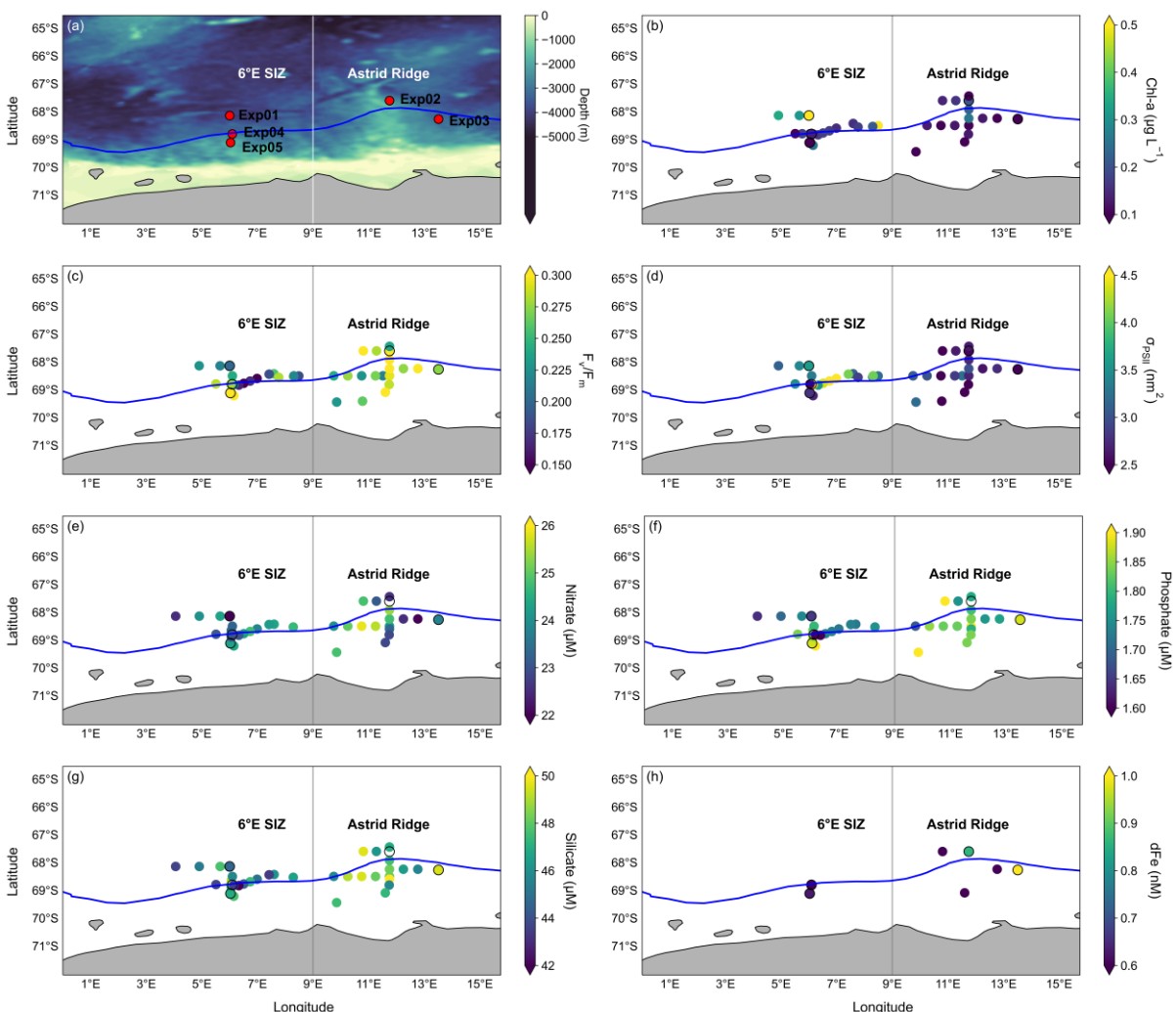

**Figure 2. Initial conditions of the study region.** Plots of (a) the overlaid bathymetry of the study region where the 6°E SIZ and Astrid Ridge region are indicated, along with the sampling locations for the incubation experiments (during March); and the associated mean initial parameters for (b) Chl-a concentrations ($\mu$g L$^{-1}$), (c) $F_v/F_m$, (d) $\sigma_{PSII}$ (nm$^2$), (e) nitrate ($\mu$M), (f) phosphate ($\mu$M), (g) silicate ($\mu$M) and (h) dFe concentration (nM). Discrete seawater samples from the underway system, surface CTD-Rosette and the Go-Flo (from initial incubations), all sampled within the study region in March, are collectively presented in (b-g) along with the average sea ice edge (concentration at 15%) for March 2019 (blue line). Plot (h) shows the dFe concentrations which were sampled at all the Go-Flo stations. All data for incubation stations are given in Table 1, and incubation stations are indicated by a black circle outline in (b-h).

140

**Table 1. Sampling location information for the incubation stations and the associated CTD-Rosette water column station numbers from the cruise (CTD cast identifier) and mean (n=3) initial parameters for the photophysiology ($F_v/F_m$ and $\sigma_{PSII}$), as well as the associated ancillary data (i.e., Chl-a concentrations, macronutrient concentrations and dFe concentrations). Cumulative photon dose and euphotic depth were calculated as defined in materials and methods. Mixed Layer Depth (MLD) was obtained from Kauko et al. (2021). The Sea Surface Layer Temperatures (SSLTs) averaged for depths 15 to 30 m were obtained from the CTD sensor. Dominant phytoplankton community composition was taken from a combination of microscopy and CHEMTAX data from Kauko et al. (2022a; 2022b). "n.d" indicates that no data was available; "±" precedes standard deviation (n=3) and '\*' denotes that a single measurement was performed.**

| | Experiment | | | | |
|---|---|---|---|---|---|
| | **Exp01** | **Exp02** | **Exp03** | **Exp04** | **Exp05** |
| **CTD identifier** | CTD53 | CTD70 | CTD83 | CTD97 | CTD105 |
| **Initiation Date** | 12/03/2019 | 17/03/2019 | 19/03/2019 | 24/03/2019 | 26/03/2019 |
| **Initiation Time (UTC)** | 08h18 | 08h33 | 19h34 | 23h26 | 09h12 |
| **Latitude (°S)** | 68.10 | 67.56 | 68.23 | 68.76 | 69.07 |
| **Longitude (°E)** | 6.00 | 11.75 | 13.51 | 6.09 | 6.03 |
| **Sunrise (UTC)** | 05h02 | 04h57 | 04h57 | 05h45 | 05h53 |
| **Sunset (UTC)** | 18h27 | 17h45 | 17h30 | 17h38 | 17h30 |
| **Cumulative photon dose (mol photons m$^{-2}$ d$^{-1}$)** | 124 | 156 | 160 | 93 | 92 |
| **MLD (m)** | 38 | 27 | 36 | 28 | 30 |
| **Euphotic depth (m)** | 31 | 50 | n.d | n.d | 53 |
| **Mean PAR in the mixed layer (µmol photons m$^{-2}$ s$^{-1}$)** | 16.65 | 109.86 | n.d | n.d | 134.08 |
| **SSLT (°C)** | -0.33 | -1.16 | -1.76 | -1.71 | -1.86 |

| | | | | | |
|---|---|---|---|---|---|
| $F_v/F_m$ | 0.20±0.01 | 0.34±0.02 | 0.35±0.01 | 0.32±0.03 | 0.28±0.01 |
| $\sigma_{PSII}$ ($nm^2$) | 3.99±0.37 | 2.72±0.08 | 2.45±0.12 | 3.13±0.54 | 2.92±0.54 |
| Chl-a ($\mu g\ L^{-1}$) | 0.73* | 0.23* | 0.02* | 0.18* | 0.14* |
| Nitrate ($\mu M$) | 22.5* | 26.2* | 25.5* | 25.8* | 25.7* |
| Phosphate ($\mu M$) | 1.67* | 1.71* | 1.69* | 1.72* | 1.75* |
| Silicate ($\mu M$) | 43* | 48* | 48* | 43* | 44* |
| dFe (nM) | n.d | 0.86±0.05 | 1.39±0.14 | 0.56±0.05 | 0.63±0.13 |
| dFe:Nitrate (nmol:µmol) | n.d | 0.03 | 0.06 | 0.02 | 0.03 |
| dFe:Phosphate (nmol:µmol) | n.d | 0.05 | 0.82 | 0.33 | 0.36 |
| Dominant phytoplankton community composition | High diatom abundance, Flagellates, dinoflagellates | Pennate diatoms and centric diatoms | Pennate diatoms and centric diatoms | Flagellates | Flagellates |

## 2.1 Underway and surface CTD seawater sampling and measurements

Underway seawater was obtained from the ship's clean seawater sampling system at ~4 m depth between incubation stations. Samples were collected for determining Chl-a concentration, macronutrient concentrations (nitrate, phosphate and silicate) and photophysiology ($F_v/F_m$ and $\sigma_{PSII}$) (Kauko et al., 2020; 2021; 2022a; 2022b; Chierici and Fransson, 2020; Singh et al., 2022). Additionally, surface seawater samples were collected using a Seabird CTD (conductivity-temperature-depth) rosette sampler and similarly analysed for Chl-a, macronutrients and photophysiology in addition to phytoplankton community composition (Kauko et al., 2020; 2021; 2022a; 2022b; Chierici and Fransson, 2020; Singh et al., 2022). Sample processing and analysis are further detailed in

section 2.4 for phytoplankton photosynthetic photophysiology and sections 2.5 – 2.10 for ancillary data. In addition, initial *in situ* conditions for the incubation experiments from CTD surface samples are detailed below in section 2.3 (Incubation set-up and sub-sampling).

## 2.2. Surface seawater sampling for incubation experiments

Seawater samples for experimental station Exp01 were collected at 20 m depth using a Watson Marlow Varmeca (MG0723) peristaltic pump connected to PTFE tubing with a 10 mm inner diameter at a flow rate of 1.6 L min$^{-1}$. All sampling tubing (peristaltic and PTFE) and 1 L Polycarbonate bottles (ThermoFisher Scientific Nalgene) were acid-washed following GEOTRACES protocols (Cutter et al., 2017). Next, inside a custom-made HEPA air-filtered Class-100 trace metal clean 'plastic bubble', that consisted of a clean, steady laminar flow hood (AirClean-600 PCR Workstation), the seawater was pumped into seven 1 L polycarbonate bottles, all this under strict trace metal clean conditions. For the other experiments (Exp02, Exp03, Exp04 and Exp05), a Teflon-lined, trace metal clean, external closure 8 L Go-Flo bottle (General Oceanics), was deployed on an aramid rope (VGP industries), using a dedicated winch and Teflon coated messenger to ~20 – 30 m depth for surface incubation seawater (i.e., for experimental stations Exp02, Exp03, Exp04 and Exp05). At each of the five experimental stations (see Fig. 2a and Table 1 for locations), seven 1 L polycarbonate bottles were filled unscreened (i.e., no large grazers were excluded from the bottles) with the incubation seawater to represent 1 x the initial sample (hereafter 'initial'), 3 x the unamended control samples (hereafter 'Control'), and 3 x iron addition samples (hereafter 'Fe'), which were spiked with 2.0 nM iron (III) chloride (FeCl$_3$ TraceCERT®; Sigma Aldrich) prepared in 2‰ HCl (30% suprapur HCl; Merck), to reach a final concentration of 2.0 nM Fe. The bottle caps of the Control and Fe samples were sealed with Parafilm™, and the bottles were double-bagged in clear polyethylene bags (ZipLoc™) to avoid sample contamination. All incubation bottle filling, spiking and sub-sampling were performed under a clean, laminar flow hood (AirClean-600 PCR Workstation), inside a makeshift HEPA air-filtered Class-100 trace metal clean bubble on-board, under strict trace metal clean conditions.

## 2.3. Incubation set-up and sub-sampling

The incubation bottles were placed inside an on-deck incubator under natural sunlight, with flowing seawater, which fluctuated with the ocean temperature, passing through the incubator to mimic *in situ* seawater temperatures. The seawater temperature was measured at the ship's intake by a thermosalinograph. Light levels inside the polycarbonate bottles were approximated using a handheld 4π PAR sensor (Biosphere QSL 2100,

Biospherical Instruments Inc.) with the Logger 2100 software. A green mesh was used to filter out a fraction of the PAR on Exp01, with the PAR approximated inside the incubator bottle being 37% of sea surface PAR, whilst the remaining experiments had no filters on the incubators, and the average PAR inside the incubator bottle corresponded to 43% PAR at the sea surface. After each 24 hr period, the incubation bottles were removed from the incubator and sub-sampled under the clean, laminar flow hood (AirClean-600 PCR Workstation), inside the makeshift HEPA air-filtered Class-100 trace metal clean plastic bubble on-board as described above in section 2.2. All incubation bottles were sub-sampled for photophysiological parameters using active Chl-a fluorescence measured through Fast Repetition Rate fluorometry (FRRf) (see section 2.4), Chl-a concentration (see section 2.5) and macronutrients (see section 2.6). A complete list of sampling locations, initial parameters for the photophysiology and ancillary data, as well as other relevant information (cumulative photon dose, MLD, euphotic depth and sea surface layer temperatures) is provided in Table 1.

## 2.4. Phytoplankton photosynthetic photophysiology

Active Chl-a fluorescence was measured with a FastOcean$^{TM}$ FRRf incorporating a FastAct$^{TM}$ laboratory system (Chelsea Technology Group), operated with the single-turnover protocol set with a flash saturation sequence (100 x 1 µs flashlets with a 2 µs interval) and a relaxation sequence (25 x 1 µs with an interval of 84 µs). The power of the excitation LED ($\lambda_{450nm}$) was adjusted between samples to saturate the observed transients following manufacturer specifications. All samples were dark acclimated for ~30 min under *in situ* temperatures prior to measurement of the photophysiological (fluorescence) parameters ($F_v/F_m$ and $\sigma_{PSII}$) (Roháček, 2002) and were each blank corrected using carefully prepared 0.2 µm filtrates (Cullen and Davis, 2003). The FRRf measurements were recorded with the FastPro8 software (v1.0.55), and post-processing analysis was done in Python 3.7, using the customized package Phytoplankton Photophysiology Utilities (Ryan-Keogh and Robinson, 2021). The fluorescence response data were fitted to the saturation phase of the biophysical model of Kolber et al. (1998), with a constant connectivity coefficient $\rho$, of 0.3 (Suggett et al., 2001) to derive $F_o$, $F_m$ and $F_v/F_m$. The sample means and the standard deviation (SD) were calculated for $F_v/F_m$ and $\sigma_{PSII}$ from each set of triplicate samples. Statistical t-tests were performed to compare the mean $F_v/F_m$ and $\sigma_{PSII}$ values between the Control and Fe samples. This was done using a Levene test to check for equal variance: if the data was of equal variance, a standard student's t-test was applied, while in the case of unequal variance, a Welch's t-test was applied. Results of the t-tests are reported as statistically significant at the 95% confidence level (p-value<0.05).

## 2.5. Chlorophyll-a (Chl-a)

A volume of 500 – 1000 mL of seawater was filtered for Chl-a extraction onto GF/F filters (nominal pore size 0.7 µm; GE Healthcare) under low vacuum pressure (ca -30 kPa). Chl-a was extracted with 100% methanol at 4°C in the dark for 24 hrs (Holm-Hansen and Riemann, 1978) and was subsequently measured on-board, using a Turner 10-AU Fluorometer (Turner Designs) which was calibrated prior to the cruise using a standard calibration curve from raw Chl-a (Sigma C6144). The uncertainty in Chl-a values was estimated as 5.5% of the measured values during an earlier campaign utilising the same method and instrument (Assmy et al., 2017).

## 2.6. Macronutrients

The seawater samples for macronutrient analysis (nitrate, phosphate and silicate) were collected in 50 mL Falcon tubes for the incubation experiments and underway samples, whereas water column samples from the CTD-Rosette were collected in 20 ml vials. All samples were preserved with 250 µL of chloroform (saturated solution with 1% ethanol for stabilization). The samples were kept cold (at 4°C in a fridge) and in the dark until post-cruise analysis was performed using a spectrophotometric method following standard procedures (Grasshoff et al., 2009) at the Institute of Marine Research, Bergen, Norway on a Skalar autoanalyzer (Gundersen et al., 2022). The analyser was calibrated using reference seawater from Ocean Scientific International Ltd. The detection limits were 0.5 µM for nitrate, 0.06 µM phosphate and 0.7 µM for silicate. The uncertainty for nitrate and silicate was <0.2% and <1% for phosphate (Gundersen et al., 2022).

## 2.7. Dissolved Fe (dFe)

Seawater samples for dFe measurements were collected from the clean Go-Flo bottles (5L General Oceanics), at seven stations in the study region (unfortunately, a dFe sample is not available for experimental station Exp01), into acid washed 125 ml low-density polyethylene (LDPE, Nalgene, ThermoScientific) sampling bottles. The LDPE bottles were acid cleaned according to the GEOTRACES protocols (Cutter et al., 2017) prior to the cruise. The dFe samples were filtered through sequential Sartorius capsule filters (0.45 and 0.2 µm pore size filtration) using acid-washed Tygon tubes inside the trace metal clean plastic bubble. During filtration, an additional HEPA air-filter cartridge (HEPA-CAP/HEPA VENT, 75 mm, Whatman) was connected to the pressure relief valve of the Go-Flo bottles to ensure that the air in contact with the sample during the filtration was clean. All samples were acidified to pH < 2 with 600 µL of ~3 M double quartz distilled ultrapure $HNO_3$ (VWR, AnalaR NORMAPUR® analytical reagent), double-bagged and stored at room temperature (> 2 years)

until analysis at Stellenbosch University (TracEx, South Africa) as described in Samanta et al. (2021) using online pre-concentration methods. Although the samples were stored for more than two years before analysis, the dFe concentration is unlikely to be affected. The long-term analyses (2017 – 2021) of GEOTRACES and Certified reference standards, which yielded consistent dFe concentrations support this conclusion (Samanta et al., 2021). All samples were measured in duplicate. The detection limit of Fe was 0.08 nM, and the precision 11% (Samanta et al., 2021).

## 2.8. Satellite Chlorophyll Data

Ocean colour data (8 days, 4 km) were obtained from the ocean colour climate change initiation (OC-CCI) (Sathyendranath et al., 2019). In order to deduce missing data, satellite-derived Chl-a values were first re-gridded to a 4 km regular grid by averaging all data points within the new pixel dimensions. Gaps in the data were filled by applying a linear interpolation scheme as defined in Racault et al. (2014). The data were smoothed by applying a moving average filter of the previous and next time step (for more details on this method see Salgado-Hernanz et al. (2019)). Two boxes were defined for the respective regions of this study and averaged to get the annual cycle of Chl-a concentration: 6°E SIZ (62 – 72°S; 0 – 9°E) and Astrid Ridge (62 – 72°S; 9 – 16°E) (Fig. 3).

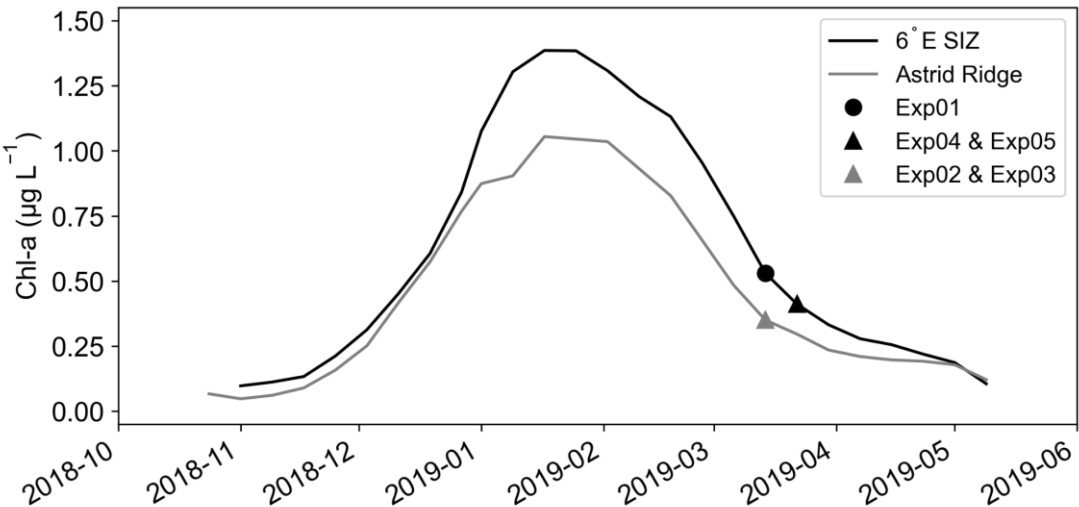

**Figure 3. Satellite chlorophyll-a (Chl-a; μg L⁻¹) data from OC-CCI from 01/10/2018 to 01/06/2019. The 6°E SIZ (62 – 72°S; 0 – 9°E) and Astrid Ridge (62 – 72°S; 9 – 16°E) were created from spatial means as indicated in the methods. The dates of the experimental set-ups are included for each region.**

### 2.9. PAR sensor data

The cumulative photon dose for each experiment (mol photons m⁻² d⁻¹) was calculated as the cumulative sum of
270 the PAR measured above the sea surface by a Biospherical Licor Chelsea PAR sensor on the ship's mast, starting from the time of experimental commencement, until experiment termination (i.e., summed over 24 hours). Values were adjusted by ~43% to account for shading within the incubator in accordance with the measured sea surface PAR inside the incubator.

### 2.10. Bathymetry data, stratification, the Mixed Layer Depth (MLD), euphotic depth and sea ice
**concentration**

The ETOPO1 bathymetry data for the study region was extracted from ("*ETOPO1, Global 1 Arc-minute Ocean Depth and Land Elevation From the US National Geophysical Data Center (NGDC),*" 2011). The degree of stratification was obtained from the Brunt-Väisälä frequency ($N^{2-}$; s⁻¹) (Millard et al., 1990), which was calculated using the seawater temperature, salinity and potential density ($\sigma$) at each experimental station. The
280 MLD for each experimental station was obtained from Kauko et al. (2021) and the respective euphotic depth was determined as the depth at which PAR is 1% of surface PAR, based on Kirk (1994). The sea ice

concentration was plotted at 15% concentration for the sea ice edge around Antarctica (Fig. 1 and Fig. 2) (Brodzik and Stewart, 2016).

## 3 Results

In previously published work from this cruise (Kauko et al., 2021), two distinct regions were identified in the DML SIZ. Both regions were visited in post-bloom conditions during the month of March (Kauko et al., 2021), but differed in the peak Chl-a concentrations, i.e., in the bloom amplitude (Fig. 3). The first region was in shallower bathymetry (2556±724 m depth, 11 – 14°E, 67 – 69°S) around Astrid Ridge (Fig. 2a). Two short-term iron addition incubation experiments, Exp02 and Exp03, were conducted in this region, north and east of Astrid Ridge, respectively (Table 1; Fig. 2a). The second region in deeper bathymetry (3042±1129 m depth, 5 – 7°E, 67 – 70°S) was located on a 6°E transect to the west of the Astrid Ridge in the open-ocean SIZ (6°E SIZ) (Fig. 2a), where the experimental station Exp01 was conducted. Despite being occupied in post-bloom conditions from a seasonal perspective (Fig. 3), Exp01 was, nonetheless, considered to represent autumn bloom conditions (Kauko et al., 2021; Moreau et al., 2023), albeit in decline, with a high Chl-a concentration (0.73 µg L$^{-1}$, Table 1). Experimental stations Exp04 and Exp05 were sampled two weeks after Exp01, which was after the seasonal bloom (Chl-a = 0.18 µg L$^{-1}$ and 0.14 µg L$^{-1}$, respectively) within the same 6°E SIZ region. We note that the starting time of each incubation was not synchronized (Table 1) and may lead to issues in interpreting photophysiological responses due to diurnal variation (Schuback et al., 2016). However, we found no distinct diurnal differences in both $F_v/F_m$ and $\sigma_{PSII}$ across the 6°E SIZ and Astrid Ridge regions (Fig. A3), with both parameters showing very little variability between local sunrise and sunset.

Here, we first describe the general conditions in these two regions (n=34) and then focus specifically on the five experimental stations. The Chl-a concentrations were lower around Astrid Ridge, ranging from 0.03 µg L$^{-1}$ to 0.26 µg L$^{-1}$ (mean 0.12±0.07 µg L$^{-1}$; n=16). Concentrations between 0.07 µg L$^{-1}$ and 1.02 µg L$^{-1}$ (mean 0.25±0.24 µg L$^{-1}$; n=18) were observed in the 6°E region of the SIZ (Fig. 2b; Table B1). The mean values of $F_v/F_m$ (Fig. 2c) were higher (p-value<0.05) at Astrid Ridge (0.28±0.04) compared to the 6°E SIZ (0.24±0.06). The 6°E SIZ showed a much larger range in $F_v/F_m$ with a minimum of 0.07 and a maximum of 0.34, whilst a narrower range in $F_v/F_m$, with a higher minimum in particular, was seen around Astrid Ridge (0.21 to 0.36). The $\sigma_{PSII}$ (Fig. 2d) was typically higher in the 6°E SIZ region, ranging from 2.48 to 5.63 nm$^2$ (mean 3.41±0.71 nm$^2$) and lower around the Astrid Ridge, 1.93 to 3.56 nm$^2$ (mean 2.66±0.37 nm$^2$).

Surface nitrate concentrations showed some spatial variability, but the mean values were similar (p-value>0.05) for the 6°E SIZ (mean 23.8±0.8 µM) and Astrid Ridge (mean 24.0±1.2 µM) (Fig. 2e). Despite a similarity in the range of phosphate concentrations observed for both the regions from 1.57 to 1.96 µM in the 6°E SIZ (mean 1.75±0.10 µM), and from 1.68 to 1.92 µM at Astrid Ridge (mean 1.82±0.06 µM), the phosphate concentrations between the regions were significantly different (p-value<0.05) (Fig. 2f). Silicate concentrations

showed a higher mean (48±1 µM, p-value<0.05) and less variability around Astrid Ridge with concentrations ranging from 46 to 52 µM, compared to a lower mean (46±2 µM) and larger range (41 to 49 µM) observed in the 6°E SIZ (Fig. 2g). Despite the limited number of dFe measurements, a wide range of surface concentrations (Fig. 2h) were evident around Astrid Ridge with concentrations as low as 0.27 nM and as high as 1.39 nM (mean 0.64±0.49 nM). Mean dFe concentrations in the 6°E SIZ were slightly lower (0.59±0.05 nM) compared

to Astrid Ridge and varied over a narrow range between 0.56 to 0.63 nM. However, it is noted that only a fraction of the dFe is bioavailable to the phytoplankton, where this fraction can vary regionally and thus influence the variability in iron stress which may not mirror the ambient concentrations (Lis et al., 2015). Furthermore, the mean PAR in the mixed layer for the 6°E SIZ was lower (29.71 µmol photons $m^{-2}$ $s^{-1}$) in comparison to the Astrid Ridge (59.37 µmol photons $m^{-2}$ $s^{-1}$).

In the following, we focus particularly on the upper ocean conditions at stations where incubation experiments were conducted (Table 1). Initial conditions in surface Chl-a ranged from high concentrations at the bloom station Exp01 (0.73 µg $L^{-1}$), to concentrations as low as 0.02 µg $L^{-1}$ at Exp03 in the Astrid Ridge. Similar to the general oceanographic conditions, both nitrate and phosphate showed very little variability between experiments, whereas silicate concentrations were slightly lower for all three stations in the 6°E SIZ (43 – 44

µM) in comparison to the Astrid Ridge (48 µM). Unfortunately, the initial dFe concentration at the bloom station Exp01 is not available, however, dFe concentrations tended to be lower at the remaining stations (Exp04 and Exp05) in the 6°E SIZ (0.56 – 0.63 nM) compared to the Astrid Ridge (0.86 – 1.39 nM) (Table 1). The cumulative photon doses over 24 hrs (Table 1; Fig. A1) were substantially different, as Exp01, Exp02 and Exp03 (124 – 160 mol photons $m^{-2}$ $d^{-1}$) had much higher doses compared to Exp04 and Exp05 (92 – 93 mol

photons $m^{-2}$ $d^{-1}$). The MLD at all experimental stations showed little variability (Kauko et al., 2020; 2021; Table 1; Fig. A2) ranging between 27 m and 38 m (mean 31±5 m). The degree of stratification, however, ranged substantially being particularly stratified at the bloom station (Exp01), with a high degree of variability in the Brunt-Väisälä frequency ($N^2$) at the MLD, and comparatively weakly stratified at Exp05, with very little variability in the profile of $N^2$ (Fig. A2). The euphotic depth ranged from 31 to 53 m at the three stations where

CTD profiles were collected during daylight hours (Table 1). Since the euphotic depth was typically deeper than

the MLD, these stations may unlikely be light-limited. However, mean PAR in the mixed layer had a broad range from 16.65 μmol photons $m^{-2}$ $s^{-1}$ (Exp01) to 134.08 μmol photons $m^{-2}$ $s^{-1}$ (Exp05), that likely reflects the degree of cloudiness (since time of day was similar), thus preventing us from making any definitive conclusions on light limitation. Although still in the negative, surface layer temperatures were warmer at the bloom station 345    Exp01 (-0.33°C) and cooler at the remaining stations (-1.16 to -1.86°C) (Table 1).

        Given the variability described above, it is anticipated that initial conditions of $F_v/F_m$ and $\sigma_{PSII}$ would vary between incubation stations (Table 1; Fig. 4). The $F_v/F_m$ was lower in the 6°E SIZ (mean 0.27±0.01) compared to Astrid Ridge (mean 0.35±0.01) and much lower at the bloom station Exp01 (0.20±0.01). The opposite was true for $\sigma_{PSII}$ with initial conditions being higher in the 6°E SIZ (mean 3.35±0.28 $nm^2$) and the 350    highest $\sigma_{PSII}$ at Exp01 (3.99±0.37 $nm^2$) with the lowest $\sigma_{PSII}$ at the Astrid Ridge (mean 2.59±0.05 $nm^2$). The differences in these initial conditions, i.e., seasonal timing and bloom amplitude, dFe surface concentrations, as well as $F_v/F_m$ and $\sigma_{PSII}$, indicate that some variability in the photophysiological response to iron addition could be anticipated. Nonetheless, despite these initial differences in conditions, very little variability was observed in the photophysiological response to iron addition (Fe) relative to the Controls (Fig. 4; Table 2). A statistical t-test 355    between Fe and Control samples confirmed this, with no significant differences (p-value>0.05) in the photophysiology ($F_v/F_m$ or $\sigma_{PSII}$) evident for any of the incubation experiments between treatments (Table 2). Similarly, no significant differences (p-value>0.05) were observed in either macronutrient or Chl–a concentrations (Table 2) between the Fe and Control incubations.

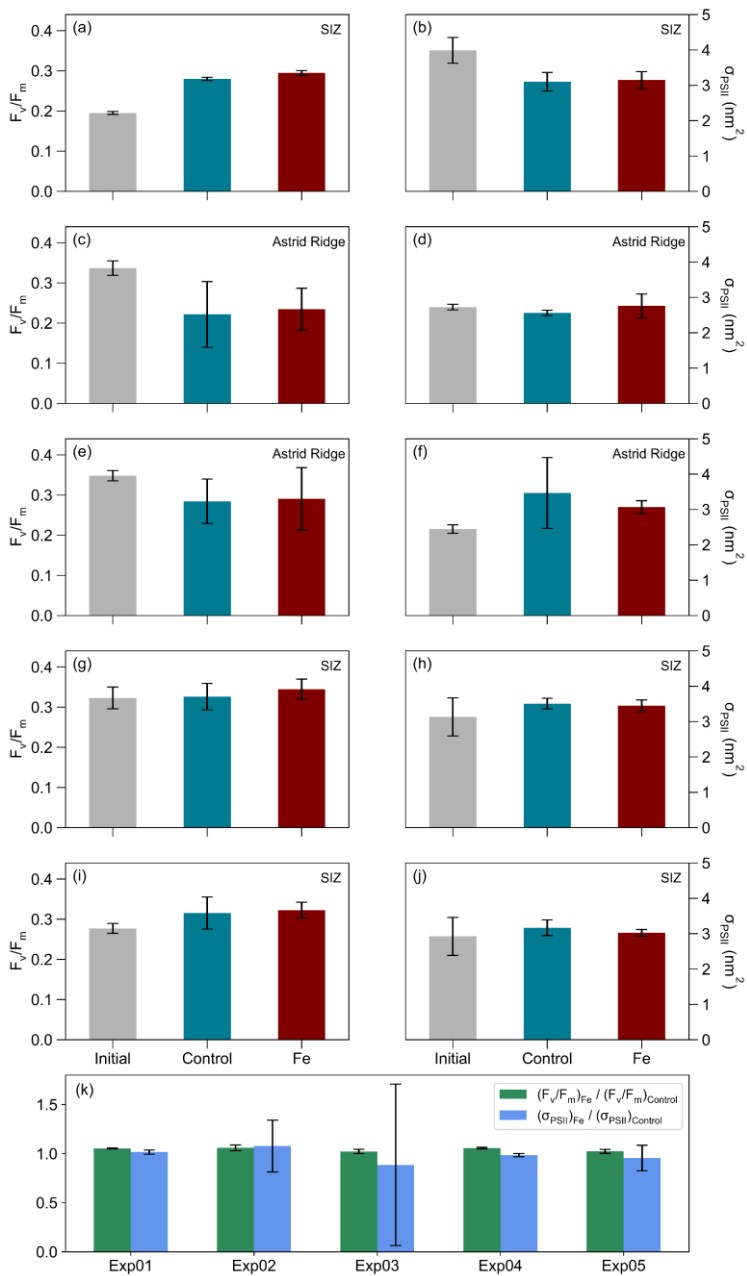

360

**Figure 4. The mean (n=3) $F_v/F_m$ and mean $\sigma_{PSII}$ (nm$^2$) from the initial, and the Control and Fe treatments, where error bars indicate standard deviations. (a,b) Exp01; (c,d) Exp02; (e,f) Exp03; (g,h) Exp04 and (i,j) Exp05, while in (k) the ratio between the Fe and Control samples for $F_v/F_m$ and $\sigma_{PSII}$ are shown for each experiment and error bars indicate standard deviations.**

**Table 2. Associated mean (± standard deviations) parameters of the Control and Fe samples measured post-incubation for $F_v/F_m$, $\sigma_{PSII}$ (nm$^2$), Chl-a concentrations (µg L$^{-1}$) and the macronutrients nitrate, phosphate and silicate (µM). "n.d" indicates that no data was available.**

| Experiment | $F_v/F_m$ | | $\sigma_{PSII}$ | | Chl-a | | Nitrate | | Phosphate | | Silicate | |
|---|---|---|---|---|---|---|---|---|---|---|---|---|
| | Control | Fe | Control | Fe | Control | Fe | Control | Fe | Control | Fe | Control | Fe |
| **Exp01** | 0.28±0.01 | 0.29±0.01 | 3.10±0.26 | 3.15±0.24 | n.d | 0.97±0.11 | 22.9±0.2 | 23.0±0.1 | 1.63±0.02 | 1.68±0.04 | 42±0 | 42±1 |
| **Exp02** | 0.22±0.08 | 0.23±0.05 | 2.56±0.07 | 2.76±0.34 | 0.19±0.02 | 0.15±0.02 | 26.4±0.2 | 27.8±0.6 | 1.66±0.02 | 1.69±0.01 | 48±0 | 47±1 |
| **Exp03** | 0.28±0.06 | 0.29±0.08 | 3.46±0.90 | 3.07±0.18 | 0.06±0.02 | 0.05±0.01 | 25.9±0.1 | 25.7±0.6 | 1.70±0.00 | 1.69±0.01 | 47±0 | 47±0 |
| **Exp04** | 0.33±0.03 | 0.34±0.03 | 3.51±0.15 | 3.45±0.16 | 0.18±0.02 | 0.17±0.01 | 26.0±0.4 | 25.7±0.6 | 1.74±0.01 | 1.73±0.01 | 44±1 | 44±0 |
| **Exp05** | 0.32±0.04 | 0.32±0.02 | 3.17±0.22 | 3.02±0.09 | 0.15±0.00 | 0.13±0.01 | 26.4±0.4 | 26.1±0.3 | 1.76±0.01 | 1.74±0.01 | 45±0 | 46±0 |

## 4 Discussion

The majority of Southern Ocean incubation studies have shown that phytoplankton are iron-limited (de Baar et al., 1990; Viljoen et al., 2018; Ryan-Keogh et al., 2017a; 2018; Browning et al., 2014a; 2014b). However, no studies, to our knowledge, have been conducted in the SIZ during autumn. Furthermore, the majority of these iron-addition incubation studies were conducted as longer-term incubations (>96 hrs). The complexity induced by longer-term nutrient addition incubations are exacerbated by artefacts that cause an isolated system to be devoid of natural factors. These natural factors include nutrient resupply and grazing which differs between the initial and incubated samples, whilst retaining only a specific sampled section from the water column as representative of the entire system (Geider and La Roche, 1994). While short-term incubations, within 24 hrs, are also an isolated system devoid of these natural factors, the impact of these factors are reduced in the shorter incubation timeframe and increased by the longer incubation timeframe. Thus, short-term incubation studies provide a sufficient period for eliciting a measurable photophysiological response (e.g., Ryan-Keogh et al., 2017a), while at the same time minimising the possibilities of artefacts in the incubation, as evidenced by the absence of any significant differences in phytoplankton biomass or nutrient concentrations between the Control samples after incubation and the initial samples before incubation. Indeed, other studies in the Southern Ocean have also reported significant changes in $F_v/F_m$ within 24 hrs following iron addition (Boyd and Abraham, 2001; Hinz et al., 2012; Browning et al., 2014a; 2014b; Ryan-Keogh et al., 2017a), suggesting that it is possible to determine rapid (<24 hrs) responses of photophysiology in iron-limited phytoplankton.

An annual time series of satellite-derived Chl-a averaged over the Astrid Ridge and 6°E SIZ region depicts the timing of the cruise relative to the seasonal cycle (Fig. 3) and clearly shows that both regional occupations were towards the end of the seasonal bloom. Therefore, it was anticipated that the region would be iron-limited and would respond favourably to iron addition. The study also covered a broad range of conditions when comparing the Astrid Ridge and 6°E SIZ regions (Fig. 2) i.e., shallower versus deeper bathymetry, lower versus higher biomass, lower versus higher dFe concentrations, lower versus higher $F_v/F_m$ and higher versus lower $\sigma_{PSII}$. Similarly, the average phytoplankton community composition between the two regions differed substantially (Kauko et al., 2022a; 2022b), where pennate diatoms (72%) and centric diatoms (56%) dominated in the Astrid Ridge region (Exp02 and Exp03), while the 6°E SIZ region consisted mostly of flagellates (Exp04 and Exp05, 45%), with the exception of Exp01 that together with flagellates had a high abundance of diatoms

(74%). Despite contrasting conditions in physics (density, stratification, cumulative photon dose, mean PAR in the mixed layer), chemistry (nitrate, silicate and dFe) and biology (Chl-a, $F_v/F_m$, $\sigma_{PSII}$ and community composition), none of the five iron incubation experiments displayed any significant differences between the Fe and the Controls for photophysiology, or for any of the ancillary parameters (Table 1 and Figs. A1 and A2). As such, iron was not considered limiting to photosynthesis at any of the autumn stations in the DML SIZ. This unexpected finding implies that despite the timing of the cruise occupation relative to the seasonal bloom termination, iron was unlikely the primary driver of the bloom's termination (Kauko et al., 2021). Coincidently, Ryan-Keogh et al. (2023) proposed a greater probability of iron limitation in spring and summer in comparison to autumn and winter, which aligns with the results of our study. Furthermore, upon evaluating the initial dFe:nitrate (nmol:μmol) and dFe:phosphate (nmol:μmol) ratios (Table 1) for the experimental stations, it is worthy to note that the dFe:nitrate ratios appear to be higher than reported values, for example, the winter-time assessment of dFe and nitrate distributions of Ellwood et al. (2008) in the South Tasman Sea of the Southern Ocean. Ellwood et al. (2008) reported a low range of dFe:nitrate ratios (0.005 – 0.018 nmol:μmol) further south from ~52°S, which corresponded with other HNLC regions that reported iron limiting conditions under low dFe:nitrate ratios (~0.01 nmol:μmol) (Ellwood et al., 2008 and references therein). Based on this evidence, the high dFe:nitrate ratios from our study (0.022 – 0.055 nmol:μmol, Table 1) indicate very little probability for an iron limitation, but rather a limitation on light and/or other trace metals such as manganese instead (Wu et al., 2019; Browning et al., 2021; Hawco et al., 2022). The observed iron concentrations and these results suggests either an internal short-term or continuous supply of dFe that prevent the bloom from exhausting a finite dFe reservoir that would otherwise be expected so late in the growing season from a stratified water column. An example of the former mechanism could be a dFe supply from remineralisation in which high bacterial abundance could serve as a proxy (Boyd et al., 2010a; Tagliabue et al., 2017; Bressac et al., 2019) based on seasonal timing of the cruise occupation (i.e., post-bloom peak in autumn). This high bacterial abundance has been observed previously by Richert et al. (2019) during spring and summer in the Amundsen Sea, who suggested high bacterial abundance as a contributing factor to sustaining and promoting phytoplankton growth in autumn beyond the spring to summer bloom season. However, the bacterial abundance observed at both the Astrid Ridge (3.8 x $10^5$ cells mL$^{-1}$) and in the Southern section of the bloom region along the 6°E transect (3.9 x $10^5$ cells mL$^{-1}$) were only slightly higher than at the bloom station Exp01 (2.6 x $10^5$ cells mL$^{-1}$) (Kauko et al., 2021). These ranges were similar to the bacterial abundance previously observed in other Southern Ocean studies (Evans and Brussaard, 2012) and during different bloom phases (Fourquez et al. 2015; Christaki et al. 2021). Conversely, the external, continuous supply of dFe may be more viable, where anomalies in the easterly

winds could drive sea-ice southwards, favouring the upwelling of iron-rich, warmer deep water as suggested by Moreau et al. (2023). In addition, Kauko et al. (2021) utilised ~20 years of satellite-derived ocean colour data to suggest that the high bloom magnitude in this region was enhanced by flow patterns in the Weddell Gyre and

455 tidal current interactions with seafloor topography enhancing primary productivity by natural fertilization. And finally, considering factors that determine the bloom end, instead of a bottom-up or micronutrient limitation (e.g., a coastal manganese limitation (Wu et al., 2019; Browning et al., 2021)), other factors such as high concentrations of krill swarms which was observed by Kauko et al. (2021) around the 6°E transect, could suggest high levels of phytoplankton grazing, particularly in the Exp01 region (Moreau et al., 2023).

Furthermore, bacteria, viral lysis, ice formation and/or wind mixing, and decreasing incident light may all be considered more important in curtailing the seasonal bloom in this particular region. Indeed, the ambient iron concentrations within the study region at the time of sampling may have been sufficient to fulfil the cellular requirements of the phytoplankton (Strzepek et al., 2011).

**5 Conclusions**

The results from this study show that although in theory it is expected that parts of the Southern Ocean are iron-limited during autumn, it is not necessarily true for the Sea-Ice Zone region surrounding Astrid Ridge and along the 6°E transect. The observed *in situ* $F_v/F_m$ and $\sigma_{PSII}$ is suggestive of efficient photophysiology, since the iron addition did not lead to increased efficiency in phytoplankton photophysiology. The primary drivers of sustained iron supply to the region in support of phytoplankton growth late in the season are being potentially provided

with both from below (i.e., vertical supply from shallow bathymetry interactions with currents, as well as upwelling of iron-rich, warmer deep water) and from within (i.e., bacterial driven remineralisation). However, further examination of these sources and the type of iron being supplied is required to confirm the dominant resupply mechanism. It is recommended that future studies in this region help to bridge the knowledge gaps by studying the varying impacts of light in tandem with iron and other trace metals which may instead be limiting

during this time of the year, with an emphasis on short-term studies to understand the photophysiological response of phytoplankton in the absence of community induced responses.

# Appendix A: Appendix figures

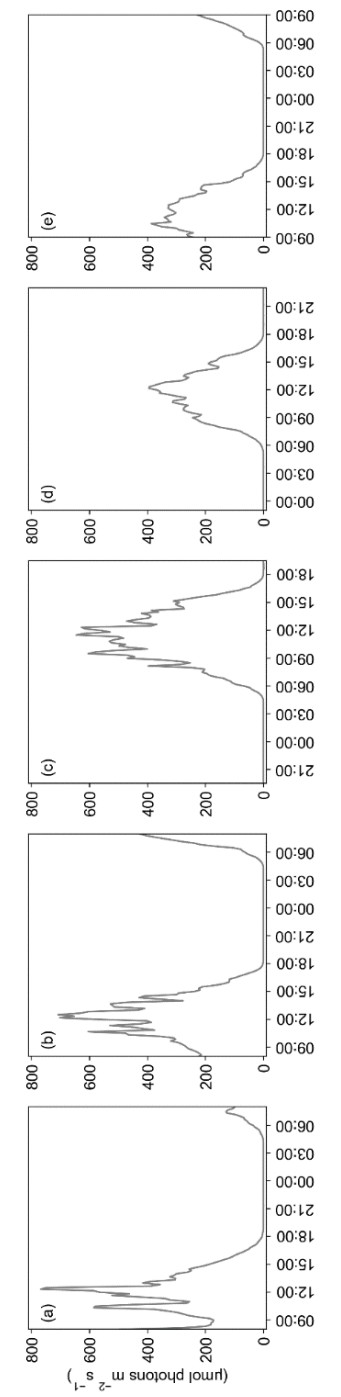

**Figure A1.** Surface PAR (μmol photons m⁻² s⁻¹) at each experimental station (a) Exp01, (b) Exp02, (c) Exp03, (d) Exp04 and (e) Exp05. Data was plotted from the time of experimental set-up until the experiment was terminated 24 hrs later.

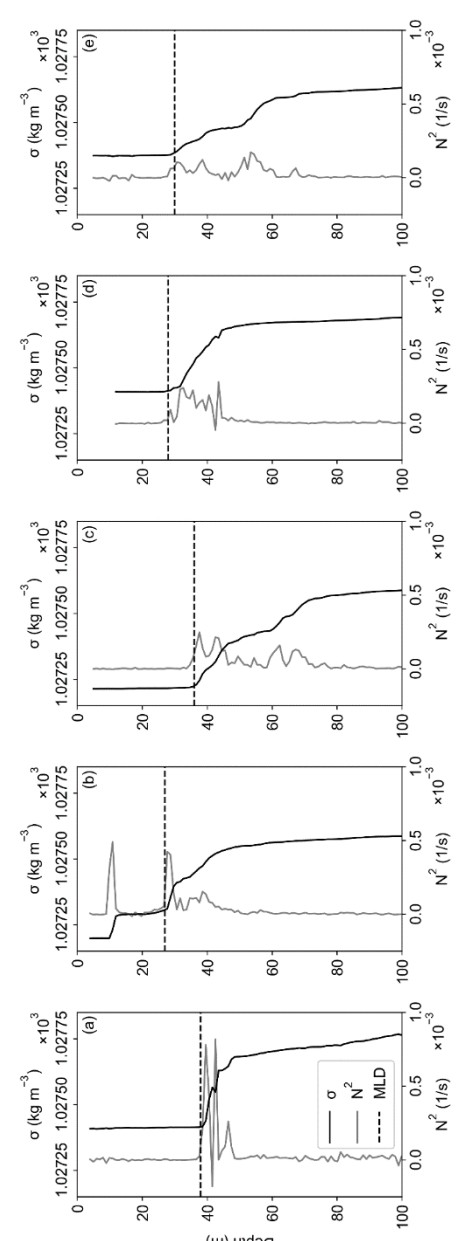

**Figure A2.** The depth profiles of density ($\sigma$; kg m⁻³) and Brunt-Väisälä frequency (N²) with the mixed layer depth (MLD; m) for experimental stations (a) Exp01, (b) Exp02, (c) Exp03, (d) Exp04 and (e) Exp05.

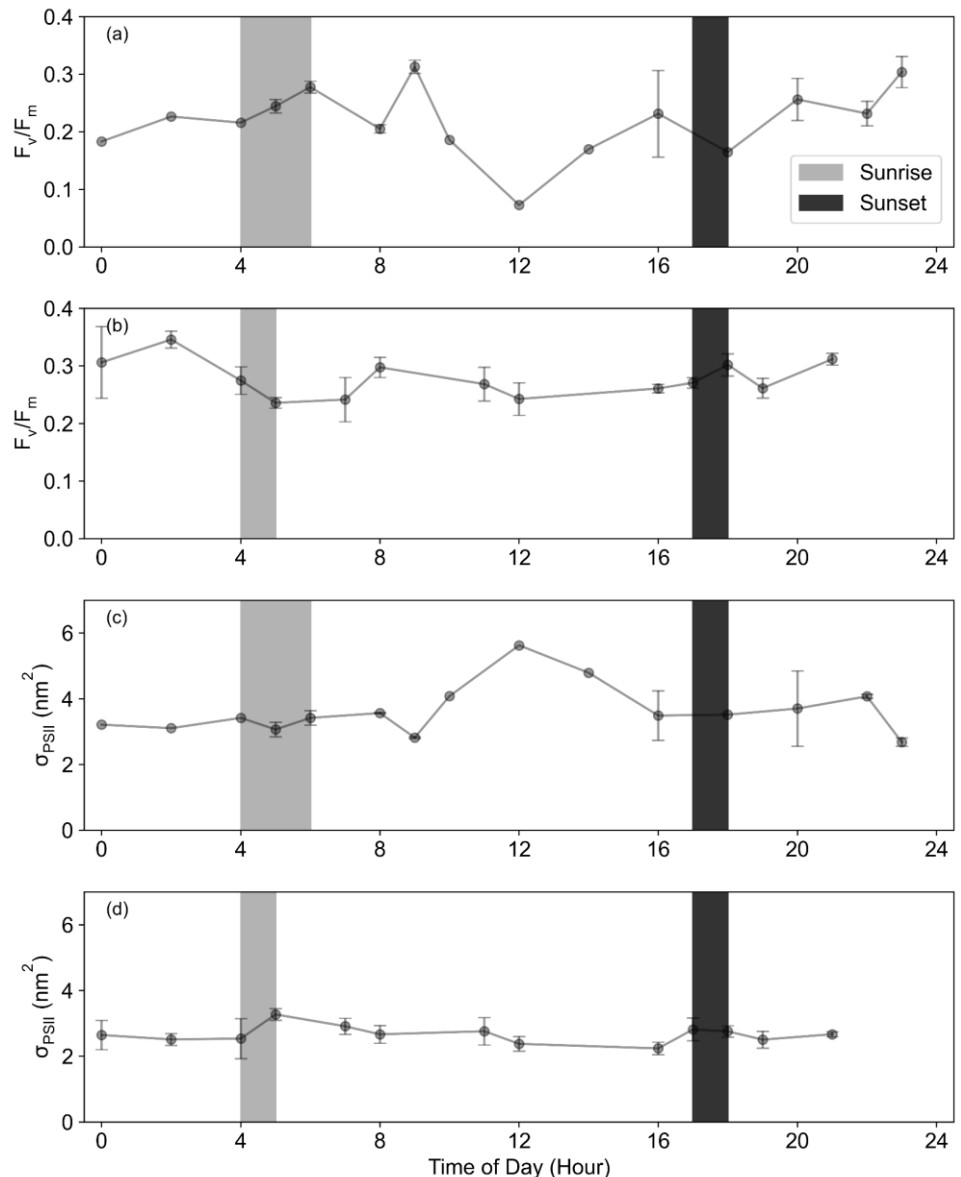

**Figure A3. The diurnal cycle of (a,b) Fv/Fm for the (a) 6°E SIZ and (b) Astrid Ridge, and of (c,d) $\sigma_{PSII}$ for the (c) 6°E SIZ and (d) Astrid Ridge, where the range of local sunrise and sunset times are indicated. Data were averaged together using the hour of the day, where error bars indicate standard deviation.**

**Table B1. Associated numbers (n), minimum, maximum and mean ($\pm$ standard deviations) parameters for the 6°E SIZ and the Astrid Ridge for Chl-a concentrations ($\mu$g L$^{-1}$), $F_v/F_m$, $\sigma_{PSII}$ (nm$^2$), macronutrients nitrate, phosphate and silicate ($\mu$M) and the dFe concentrations (nM).**

| | | Chl-a | $F_v/F_m$ | $\sigma_{PSII}$ | Nitrate | Phosphate | Silicate | dFe |
|---|---|---|---|---|---|---|---|---|
| **6°E SIZ** | **min** | 0.07 | 0.07 | 2.48 | 21.8 | 1.57 | 41 | 0.57 |
| | **max** | 1.02 | 0.34 | 5.63 | 24.8 | 1.96 | 49 | 0.63 |
| | **mean±SD** | 0.25±0.24 | 0.24±0.06 | 3.41±0.71 | 23.8±0.8 | 1.75±0.10 | 45±2 | 0.59±0.05 |
| | **n** | 18 | 33 | 33 | 21 | 21 | 21 | 2 |
| **Astrid Ridge** | **min** | 0.03 | 0.21 | 1.93 | 21.8 | 1.68 | 46 | 0.27 |
| | **max** | 0.26 | 0.36 | 3.56 | 25.9 | 1.92 | 52 | 1.39 |
| | **mean±SD** | 0.12±0.07 | 0.28±0.04 | 2.66±0.37 | 24.0±1.2 | 1.82±0.06 | 48± | 0.64±0.49 |
| | **n** | 16 | 55 | 55 | 17 | 17 | 17 | 5 |

## Data availability

All datasets on the underway samples (Chlorophyll-a, photophysiology ($F_v/F_m$ and $\sigma_{PSII}$) and nutrients (nitrate, phosphate and silicate)), as well as the incubation data which appear in this paper are available on Zenodo <Photophysiological response of autumn phytoplankton in the Antarctic Sea-Ice Zone | Zenodo>; CTD-Rosette surface photophysiology data and the surface iron data from the Go-Flo can also be found at this link. Full datasets for the other CTD-Rosette water column data are available at the Norwegian Polar Data Centre, Norwegian Polar Institute, https://data.npolar.no/dataset (Chlorophyll-a and Mixed Layer Depth) and Norwegian Marine Data Centre, https://doi.org/10.21335/NMDC-1503664923 (nutrients).

## Author contributions

TJRK conceptualized the study. AS and TJRK collected the data and performed the data analysis. MVA, NS and AS conducted the trace metal clean water collection and sampling. SM planned the general biological sampling of the cruise. SM and HMK assisted in collecting and analysing the chlorophyll-a data and all the CTD-Rosette samples. SS and AS analysed the dissolved iron samples and SS performed the data analysis and ANR guided the analysis of the dissolved iron samples. AF PI for the SANOCEAN and SOPHY-CO2 project and was involved in cruise planning of the water column sampling onboard the cruise. MC was responsible for water column collection and analyses of all the nutrient samples obtained from the underway, water column and incubation experiments. TNM supported preparations for the iron addition incubations. AS and TJRK produced the figures. AS wrote the initial manuscript. TJRK, SF, SJT and AS contributed to the study design, interpretation of the results and writing of the manuscript. All authors contributed to commenting on the manuscript.

## Competing interests

The authors declare that they have no conflict of interest.

## Acknowledgements

We would like to acknowledge the support and assistance of the captain and crew of the R/V Kronprins Haakon, along with all the participants on the DML2019702 Ecosystem research cruise. Thank you to Agneta Fransson (NPI) and Sandy Thomalla (CSIR) as the PIs of the SANOCEAN project SOPHY-CO2.

**Financial support**

The research expedition, Dronning Maud Land Ecosystem cruise 2019, and the research conducted on the RV Kronprins Haakon were part of a South African – Norwegian collaboration (SANOCEAN SOPHY-CO2), funded by the National Research Foundation (NRF), South Africa (grant UID 118715); the Research Council of Norway (RCN) project number 288370; the Norwegian Polar Institute (NPI), as well as additional financial support from the Norwegian Ministry of Foreign Affairs. AS, SJT and TJRK were supported through the CSIR's Southern Ocean and Carbon Climate Observatory (SOCCO)

Programme (http://socco.org.za/) funded by the Department of Science and Innovation (DST/CON 0182/2017) and the CSIR's Parliamentary Grant. We would like to acknowledge funding received from a number of grants from the National Research Foundation, South Africa, grant numbers 110731 (SF), 91313 (SF), 118751 (SJT) and 110729 (SJT).

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
