# Peer review of "Absence of photophysiological response to iron addition in autumn phytoplankton in the Antarctic Sea-Ice Zone"

_Biogeosciences, 2022_

## Author Comment (AC1)

**Photophysiological response of autumn phytoplankton in the Antarctic Sea-Ice Zone**

**"Absence of photophysiological response to iron addition in autumn phytoplankton in the Antarctic Sea-Ice Zone"**

Singh et al.

**RC2**: **'Comment on bg-2022-245', Anonymous Referee #2, 06 Mar 2023**

**Citation**: https://doi.org/10.5194/bg-2022-245-RC2

Singh et al examine the photophysiological response of phytoplankton communities during autumn in the Southern Ocean via iron addition incubations. No significant differences were observed in Fv/Fm and σ PSII and the authors conclude that there was not iron-limitation at these times and locations. I commend the authors for presenting what some would consider "negative results." The data are clearly presented and methods are described in detail. Please see my minor comments below.

**Note from Authors:** We thank the reviewer for their appreciation and encouragement for presenting our data, as well as the constructive feedback and suggestions that will assist in refining our manuscript. Please find below our specific responses as well as indicated changes made to the manuscript.

Please note that, as per the suggestion of Reviewer #1, we have considered changing the manuscript title as follows:
"Absence of photophysiological response to iron addition in autumn phytoplankton in the Antarctic Sea-Ice Zone"

1. **Abstract (Lines 24-26) and Line 399** - The authors state that this study confirms that the phytoplankton communities "were not iron limited and…ambient iron concentrations were sufficient." I suggest that the authors rephrase these sentences to reduce their high confidence in their assessment that there was no iron limitation although I agree with the authors that these results suggest that Fe was not the sole limiting nutrient.

**Authors' response:**
**Original:**
"The photophysiological response of phytoplankton to iron addition, measured through the photosynthetic efficiency and the absorption cross-section for photosystem II, showed no significant responses. This confirms that phytoplankton were not iron-limited at the time and that ambient iron concentrations were sufficient to fulfill the cellular requirements."

This has been changed to read as follows, in addition to a change in the title, as suggested by reviewer 1:

**Modified:**
"Contrary to expectation, the photophysiological response of phytoplankton to iron addition,

measured through the photosynthetic efficiency and the absorption cross-section for photosystem II, showed no significant responses. It is thus proposed that the autumn phytoplankton in the SIZ exhibited a lack of an iron limitation at the time of sampling, and that ambient iron concentrations may have been sufficient to fulfill their cellular requirements."

2. An alternative explanation is that there is Fe-Mn colimitation. Ratios of dMn to dFe relatively close to the study region suggest that Mn limitation or Fe-Mn limitation is possible (Browning et al. Nature Communications 2021 Supplementary Fig 5). As Mn is critical for PSII, the photophysiology results presented here could be influenced by Mn-Fe colimitation. The authors very briefly hint at this at the end of the conclusion (line 430).

**Authors' response:**
We appreciate the reviewer's suggestion to look into the Fe-Mn co-limitation. We agree that Mn is critical for PSII, and there could be a potential Mn-Fe co-limitation in the Sea-Ice Zone close to Dronning Maud Land, based on the study by Browning et al. (2021) who found low coastal dMn concentrations towards the west of our study region.

However, our intention was not to explicitly claim or suggest the limitation of any specific trace metals but encourage an interest among the community to consider this as an option when planning future campaigns in the Southern Ocean, as more seasonal studies other than in summer is needed, particularly close to the sea-ice edge of the Antarctic coast. Our concluding remarks thus strive to highlight future objectives and aims which can be achieved in this study region. Nevertheless, we have removed the specific part mentioning manganese as an example in the conclusion, so as to avoid confusion:

"It is recommended that future studies in this region help to bridge the knowledge gaps by studying the varying impacts of light in tandem with iron and other trace metals which may instead be limiting during this time of the year, with an emphasis on short-term studies to understand the photophysiological response of phytoplankton in the absence of community induced responses."

Consequently, we have included a sentence in the final paragraph of the discussion suggesting the possibility of other limitations in the region, such as Mn when listing the possible contributions to high iron concentrations (also included in the comment response #4 on high dFe:nitrate ratios below and suggestions from Reviewer #3):

"And finally, considering factors that determine the bloom end, it may not be confined to a bottom-up limitation or the possibilities of light and/or other micronutrients such as manganese instead being limiting in this sea-ice region which is close to the coast of Antarctica (Browning et al., 2021)."

**Reference:**
Browning, T.J., Achterberg, E.P., Engel, A., and Mawji, E.: Manganese co-limitation of phytoplankton growth and major nutrient drawdown in the Southern Ocean. Nat. Commun. 12, 1–9. https://doi.org/10.1038/s41467-021-21122-6, 2021.

3. Also, much of the dFe is < 1 nM (lines 279-287), and only a fraction of dFe is bioavailable, which should also be mentioned in the results/discussion.

Thank you, we have made the changes as follows:
**Original:**
"Silicate concentrations showed a higher mean (48±1 µM) and less variability around Astrid Ridge with concentrations ranging from 46 to 52 µM, compared to a lower mean (46±2 µM) and larger range (41 to 49 µM) observed in the 6°E SIZ (Fig. 1g). Despite the limited number of dFe measurements, a wide range of surface concentrations (Fig. 1h) were evident around Astrid Ridge with concentrations as low as 0.27 nM and as high as 1.39 nM (mean 0.64±0.49 nM). Mean dFe concentrations in the 6°E SIZ were slightly lower (0.59±0.05 nM) compared to Astrid Ridge and varied over a narrow range between 0.56 to 0.63 nM. Furthermore, the mean PAR in the mixed layer for the 6°E SIZ was lower (29.71 µmol photons m$^{-2}$ s$^{-1}$) in comparison to the Astrid Ridge (59.37 µmol photons m$^{-2}$ s$^{-1}$)."

**Modified:**
"Silicate concentrations showed a higher mean (48±1 µM) and less variability around Astrid Ridge with concentrations ranging from 46 to 52 µM, compared to a lower mean (46±2 µM) and larger range (41 to 49 µM) observed in the 6°E SIZ (Fig. 1g). Despite the limited number of dFe measurements, a wide range of surface concentrations (Fig. 1h) were evident around Astrid Ridge with concentrations as low as 0.27 nM and as high as 1.39 nM (mean 0.64±0.49 nM). Mean dFe concentrations in the 6°E SIZ were slightly lower (0.59±0.05 nM) compared to Astrid Ridge and varied over a narrow range between 0.56 to 0.63 nM. However, it is noted that only a fraction of the dFe is bioavailable to the phytoplankton, where this fraction can vary regionally and thus influence the variability in iron stress which may not mirror the ambient concentrations (Lis et al., 2015). Furthermore, the mean PAR in the mixed layer for the 6°E SIZ was lower (29.71 µmol photons m$^{-2}$ s$^{-1}$) in comparison to the Astrid Ridge (59.37 µmol photons m$^{-2}$ s$^{-1}$)."

**Reference:**
Lis, H., Shaked, Y., Kranzler, C., Keren, N., and Morel, F.M.: Iron bioavailability to phytoplankton: an empirical approach. The ISME journal, 9(4), 1003-1013. https://doi.org/10.1038/ismej.2014.199, 2015.
https://www.nature.com/articles/ismej2014199

4. It may also be useful to report the range in dFe:NO3 (nmol:umol) which appear to be quite high so it is also surprising that Fe did not have an effect. Again, I largely agree with the authors' conclusions; however, I believe some altered wording and added discussion of potential Fe-Mn colimitation is warranted.

**Authors' response:**
We appreciated the advice of the reviewer. Indeed, we are aware that the iron ratio to both macronutrients nitrate and phosphate render high values for each experiment (please see below).

| Exp | dFe:nitrate (nmol:µmol) | dFe:phosphate (nmol:µmol) |
|---|---|---|
| Exp01 | n.d | n.d |
| Exp02 | 0.86 : 26.2 = 0.033 | 0.86 : 1.71 = 0.50 |
| Exp03 | 1.39 : 25.5 = 0.055 | 1.39 : 1.69 = 0.82 |
| Exp04 | 0.56 : 25.8 = 0.022 | 0.56 : 1.72 = 0.33 |
| Exp05 | 0.63 : 25.7 = 0.025 | 0.63 : 1.75 = 0.36 |

In the publication of Ellwood et al. (2008), during winter-time in the South Tasman Sea of the Southern Ocean, it is reported that the dFe:nitrate ratios (as Fe:NO3) in Figure 3 exhibited a decrease in surface concentrations southward (~0.005 nmol:µmol) from 52 - 53°S. The experiments reported in our manuscript were conducted further south, between 68.56° - 69.07°S. Furthermore, Ellwood et al. (2008) report that low Fe:NO3 ratios in the south (0.005–0.018 nmol:µmol) corresponded with other HNLC regions which reported iron limiting conditions under low Fe:NO3 ratios (~0.01 nmol:µmol). Thus, we agree that the dFe:nitrate ratios in our study do indeed appear to be high; however, we disagree that it is surprising that iron addition did not have any effect on the phytoplankton, as they were not iron-limited from our results.

Moreover, we had intended to establish a proxy for potential iron limitation by assessing the uptake ratio between iron and PO4 as the tracer Fe* (Parekh et al., 2005; Rijkenberg et al., 2018). The Fe* is defined as the difference between the dFe concentration and the PO4 concentration, multiplied by a dFe:PO4 ratio (Twining et al., 2014), and is used to quantify the extent of iron limitation in the water mass. Negative Fe* values for the surface waters would thus suggest the potential for iron limitation (Parekh et al., 2005; Rijkenberg et al., 2018).

Using the ratio estimated for iron-limited Southern Ocean species (0.18 mol.mol$^{-1}$; Strzepek et al., 2011), no negative Fe* values were obtained for any of the 4 experiments (Exp01 did not have a dFe value), which suggests that there was no iron deficiency in respect to PO4 as could be expected a priori on those values. Yet, this uptake ratio does have several assumptions (most of these ratios represent laboratory conditions for single species cultures (Strzepek et al., 2011)), and by including other ratios (0.47 (Parekh et al., 2005) and 0.56 (Twining et al., 2014)), some of the experimental stations would have negative Fe* values and thus suggest an iron limitation. Given this ambiguity, we have refrained from including the Fe* values in the discussion.

However, we have included the above table depicting the dFe:nitrate and dFe:phosphate ratios into Table 1 in our manuscript and we include the following sentence in the discussion (rewritten in the paragraph from line 402 onward, taking into consideration the suggestions made by Reviewer #3) which compares these values and the interpretation thereof as described above:

"Furthermore, upon evaluating the initial dFe:nitrate (nmol:µmol) and dFe:phosphate (nmol:µmol) ratios (Table 1) for the experimental stations, it is worthy to note that the dFe:nitrate ratios appear to be higher than reported values, where for example, the winter-time assessment of dFe and nitrate distributions of Ellwood et al. (2008) in the South Tasman Sea of the Southern Ocean. Ellwood et al. (2008) reported a low range of dFe:nitrate ratios (0.005–0.018 nmol:µmol) further south from ~52°S, which corresponded with other HNLC regions which reported iron limiting conditions under low dFe:nitrate ratios (~0.01 nmol:µmol) (Ellwood et al., 2008 and references therein). Based on this evidence, the high dFe:nitrate ratios from our study indicate very little probability for an iron limitation, but rather a limitation on light and/or other trace metals such as manganese instead (Browning et al., 2021)."

**References:**
Ellwood, M.J., Boyd, P.W., and Sutton, P.: Winter-time dissolved iron and nutrient distributions in the Subantarctic Zone from 40–52S; 155–160E. Geophysical Research Letters, 35(11). https://doi.org/10.1029/2008GL033699, 2008.

Parekh, P., Follows, M.J., and Boyle, E.A.: Decoupling of iron and phosphate in the global ocean. Global Biogeochem. Cycles 19, 1–16. https://doi.org/10.1029/2004GB002280, 2005.

Rijkenberg, M.J., Slagter, H.A., Rutgers van der Loeff, M., Van Ooijen, J., and Gerringa, L.J.: Dissolved Fe in the deep and upper Arctic Ocean with a focus on Fe limitation in the Nansen Basin. Frontiers in Marine Science, 5, 88. https://doi.org/10.3389/fmars.2018.00088, 2018.

Twining, B.S., Nodder, S.D., King, A.L., Hutchins, D.A., LeCleir, G.R., DeBruyn, J.M., Maas, E.W., Vogt, S., Wilhelm, S.W., and Boyd, P.W.: Differential remineralization of major and trace elements in sinking diatoms. Limnol. Oceanogr. 59, 689–704. http://dx.doi.org/10.4319/lo.2014.59.3.0689, 2014.

Strzepek, R.F., Maldonado, M.T., Hunter, K.A., Frew, R.D., and Boyd, P.W.: Adaptive strategies by Southern Ocean phytoplankton to lessen iron limitation: Uptake of organically complexed iron and reduced cellular iron requirements. Limnology and Oceanography, 56(6), 1983-2002. https://doi.org/10.4319/lo.2011.56.6.1983, 2011.

5. Line 424 – The authors state that they observed "high Fv/Fm" although I would consider many values to be relatively low (< 0.3). I suggest changing this sentence.

**Authors' response:**
Thank you, we have made the changes as follows:
**Original:**
"The results from this study show that although in theory it is expected that parts of the Southern Ocean are iron-limited during autumn, it is not necessarily true for the Sea-Ice Zone region surrounding Astrid Ridge and along the 6°E transect, where high Fv/Fm and $\sigma$PSII, i.e. efficient photophysiology was observed in situ, and where iron addition did not lead to more efficient photophysiology."

**Modified:**
"The results from this study show that although in theory it is expected that parts of the

Southern Ocean are iron-limited during autumn, it is not necessarily true for the Sea-Ice Zone region surrounding Astrid Ridge and along the 6°E transect. The observed *in situ* Fv/Fm and $\sigma$PSII is suggestive of efficient photophysiology, and where the iron addition did not lead to increased efficiency in phytoplankton photophysiology.

6. The recent paper in Science by the corresponding author here (Ryan-Keogh et al 2023) seems relevant to include in the Discussion. Specifically, its stated that irradiance normalized NPQ is higher in Spring/Summer compared to Fall/Winter which aligns with the results in these studies.

**Authors' response:**
We thank the reviewer for suggesting the addition of the recent publication, where the conclusions of Ryan-Keogh et al. (2023) do indeed align with our conclusion that the phytoplankton studied in our manuscript was not iron-limited. The iron limitations seen in spring and summer are much greater than that seen in autumn and winter (Ryan-Keogh et al., 2023).

**Original:**
"As such, iron was not considered limiting to photosynthesis at any of the autumn stations in the DML SIZ. This was unexpected and implies that despite the timing of the cruise occupation relative to the seasonal bloom termination, iron was unlikely the primary driver of the bloom's ending."

**Modified:**
"As such, iron was not considered limiting to photosynthesis at any of the autumn stations in the DML SIZ. This unexpected finding implies that despite the timing of the cruise occupation relative to the seasonal bloom termination, iron was unlikely the primary driver of the bloom's termination (Kauko et al., 2021). Coincidentally, a recent study by Ryan-Keogh et al. (2023) proposed a greater probability of iron limitation in spring and summer in comparison to autumn and winter, which aligns with the results of our study."

**Reference:**
Ryan-Keogh, T.J., Thomalla, S.J., Monteiro, P.M., and Tagliabue, A.: Multidecadal trend of increasing iron stress in Southern Ocean phytoplankton. Science, 379(6634), 834-840. https://doi.org/10.1126/science.abl5237, 2023.

---

## Author Comment (AC2)

**Photophysiological response of autumn phytoplankton in the Antarctic Sea-Ice Zone**

**"Absence of photophysiological response to iron addition in autumn phytoplankton in the Antarctic Sea-Ice Zone"**

Singh et al.

**RC3**: **'Comment on bg-2022-245', Anonymous Referee #3, 11 Mar 2023**
**Citation**: https://doi.org/10.5194/bg-2022-245-RC3

The study by Singh et al. measure the photophysiological responses of phytoplankton communities following iron addition by looking at Fv/Fm and σ PSII variables. The study was conducted from late summer to mid-autumn in the Antarctic Sea-Ice Zone along the 6°E. Overall, no significant differences were observed between control and treatment (+Fe) and the authors concluded that dFe concentrations were not limiting for phytoplankton growth and thereby does not explain the decline of the bloom. Like reviewer 2, I agree that what could be considered negative outcomes are important, but I don't think there are enough measures or data here. I would recommend that this data set be merged with other data from the same expedition. While these data are worthy of publication, these days it is almost anecdotal to measure Fv/Fm and incubate with Fe addition during an oceanographic cruise and this type of data is usually presented as part of a broad set of metrics or analyses (omics, absorption experiment, cell abundance, etc etc). Moreover, only one depth was sampled at each o few locations (5 in total). I honestly wonder if there is enough data for a full publication.

Still, I suggest minor revisions because the data are well presented, the manuscript is well written and pleasant to read as it is. I prefer to leave the final decision on whether to publish in Biogeosciences to the editor.

**Note from Authors:** We thank the reviewer for their appreciation of our presented data, as well as the constructive feedback and suggestions that will assist in refining our manuscript.

We recognize that the data presented in this manuscript are indeed limited. Nonetheless, we consider the data to be valuable in their own right for the following reasons:

1. Emphasizing the importance of understanding seasonal limitations on phytoplankton productivity in the Southern Ocean
2. The limitation of data such as these in the vast Southern Ocean where none of the published works are experiments specifically in autumn and within this particular region
3. The results in our manuscript are in some ways contrary to what may have been expected for this season
4. We discuss our result in the context of broader understanding and literature of previous work to substantiate our interpretation despite the limited data

Please find below our specific responses, as well as the inclusion of changes made to the manuscript.

Please note that, as per the suggestion of Reviewer #1, we have considered changing the manuscript title as follows:
"Absence of photophysiological response to iron addition in autumn phytoplankton in the Antarctic Sea-Ice Zone".

More specific comments here below:

**Abstract**

1. "To better understand the constraints on productivity". « constraints » sounds a bit vague.
2. Iron addition – please specify in the abstract the nature of the addition. Inorganic Fe … 2nM…

**Authors' response:**
Thank you, we have made the changes as follows:

**Original:**
"To better understand the constraints on productivity in the Antarctic Sea-Ice Zone (SIZ), the photophysiological response of phytoplankton to iron addition was investigated during autumn along the Antarctic coast off Dronning Maud Land."

**Modified:**
"To better understand the potential limitations on productivity in the Antarctic Sea-Ice Zone (SIZ), the photophysiological response of phytoplankton to iron addition (2.0 nM FeCl₃) was investigated during autumn along the Antarctic coast off Dronning Maud Land."

**Introduction**

3. In general, I think the introduction would gain strength by citing more recent work. While I agree that one should acknowledge the work of the pioneers, one also expects to read a state of the art on the subject.

**Authors' response:**
We thank the reviewer for suggesting including more recent publications to strengthen our literature review. We also note that Reviewer #1 had suggested to include more original work and publications from the period of 1988 - 2005 regarding incubation experiments and iron limitation. Statements in the introduction have thus been adjusted to include more literature as suggested.

**Modified:**

Examples:
"In spring, phytoplankton blooms are initiated when there is sufficient light, driven by a shoaling of the mixed layer (Moore and Abbott, 2002; Thomalla et al., 2011) as well as retreating sea-ice (Taylor et al., 2013) to support phytoplankton growth under nutrient replete conditions (Swart et al., 2015; de Baar et al., 1990; Hauck et al., 2015; Martin et al., 1990)."

"Active Chl-a fluorescence is a key indicator of the photophysiological state of phytoplankton (Hughes et al., 2018; Brown et al., 2019; Schuback et al., 2021) and provides a powerful tool for evaluating the photophysiological response of phytoplankton to iron addition, i.e. by measuring the photosynthetic efficiency, Fv/Fm, and the absorption cross-section of photosystem II, σPSII (Geider, 1993; Geider and La Roche, 1994; Kolber et al., 1988; 1994; Hughes et al., 2018)."

"For instance, iron limitation is commonly associated with the pelagic waters of the Southern Ocean (Mitchell et al., 1991), where summer dissolved iron (dFe) concentrations in surface waters are typically <0.5 nM (Sedwick et al., 1999; Coale et al., 1999; Vink and Measures, 2001; Klunder et al., 2011); however, there are a number of regional exceptions."

We list below additional publications (and references therein) which will be included in our introduction accordingly.

**References:**

[revised manuscript text omitted]

6. Line 66: What about grazing ?

**Authors' response:**
If we are quoting the correct sentence from line 66 below, then we would like to point out that the sentence that follows this, continues to highlight grazing, among other factors that may result in bloom termination.
**Original:**
"Blooms typically subside when nutrients such as iron are depleted in late summer or early autumn (Tagliabue et al., 2014; Soppa et al., 2016)."

**Clarified:**
"Blooms typically subside when nutrients such as iron are depleted in late summer or early autumn (Tagliabue et al., 2014; Soppa et al., 2016). Grazing (Lancelot et al., 1993; Moreau et al., 2020; Kauko et al., 2021), bacteria and viruses (Biggs et al., 2021) may also accelerate the blooms' demise."

7. Line 77: Work by Lannuzel ?

**Authors' response:**
Thank you for pointing out the omission of the reference.
**Original:**
"There is thus minimal information on the impact of iron addition in the Sea-Ice Zone (SIZ) in autumn, when iron concentrations are expected to be low (Tagliabue et al., 2014)."

**Modified:**
"There is thus minimal information on the impact of iron addition in the Sea-Ice Zone (SIZ) in autumn, when iron concentrations are expected to be low (Tagliabue et al., 2014; Lannuzel et al., 2016)."

**Reference:**
Lannuzel, D., Vancoppenolle, M., van Der Merwe, P., De Jong, J., Meiners, K.M., Grotti, M., Nishioka, J., and Schoemann, V.: Iron in sea ice: Review and new insightsIron in sea ice: Review and new insights. Elementa: Science of the Anthropocene, 4. https://doi.org/10.12952/journal.elementa.000130, 2016. https://online.ucpress.edu/elementa/article/doi/10.12952/journal.elementa.000130/112863/Iron-in-sea-ice-Review-and-new-insightsIron-in-sea

8. I like the way differences between short and long term experiment provide different kind of information. It is rarely explained.

**Authors' response:**
Thank you kindly for appreciating our efforts in this regard.

9. Line 101: please specify what differences in physiology (Fv/Fm?) for this statement

**Authors' response:**
Thank you for suggesting the clarification needed. We have made the changes as follows:
**Original:**
"In this paper, we opted instead for short-term (24 hr) incubation experiments to isolate changes in photophysiology."

**Modified:**
"In this paper, we opted instead for short-term (24 hr) incubation experiments to isolate changes in photophysiology, i.e., $F_v/F_m$ and $\sigma_{PSII}$."

**Materials and Methods**

10. Line 115: what and where are the ancillary data? Please specify.

**Authors' response:**
We have attempted to clarify the text as per below:
**Original:**
"Ancillary data from surface water samples provide information on the regional conditions surrounding the five incubation experiments at the time of the cruise."

**Modified:**
"Ancillary data (i.e., Chl-a concentrations, macronutrient concentrations and dFe concentrations) from surface water samples provide information on the regional conditions surrounding the five incubation experiments at the time of the cruise."

11. Figure 1 – from 28$^{th}$ Feb to 10 April "initials conditions" between stations cannot be compared. It would be helpful to see the track of the vessel to visually assess what stations was sampled late summer/early autumn versus mid-autumn (April).

**Authors' response:**
Thank you for pointing this out, as we now see that the presentation can easily be misunderstood. We did in fact only compare stations within the 6°E SIZ region (62 - 72°S; 0 - 9°E) and the Astrid Ridge region (62 - 72°S; 9 - 16°E), which were conducted during the month of March (same period as our experiments). The cruise began in Punta Arenas, Chile, on 28th February 2019 and ended on the 10th of April 2019 in Cape Town, South Africa. However, both the incubation experiments presented (kindly refer to Table 1) and the initial conditions assessed for the two regions in this study, were both measured during March in the same vicinity (i.e. the general SIZ region).

To clarify in the text, we have edited the beginning of the Results section as follows:
"In previously published work from this cruise (Kauko et al., 2021), two distinct regions were identified in the DML SIZ. Both regions were visited in post-bloom conditions during the month of March (Kauko et al., 2021), but differed in the peak Chl-a concentrations, i.e. in the bloom amplitude (Fig. 2)."

Furthermore, we have opted to plot a separate, larger map of the study region which includes the cruise track, parts of South America and South Africa (where the cruise began and ended), the five experimental locations, bathymetry of the entire map and the sea ice extent (plotted at 15% sea ice concentration).

Potential Figure caption:
"Figure 1. Map of the general study region and cruise track of the DML2019702 cruise that began in Punta Arenas, Chile, on 28th February 2019 and ended on the 10th of April 2019 in Cape Town, South Africa. The overlaid bathymetry of the study region is shown where the 6°E SIZ and Astrid Ridge region are indicated and the respective incubation experiments sampling locations (in March) are indicated close to the coast off Dronning Maud Land in the Sea-Ice Zone (SIZ). The sea-ice extent is plotted at 15% sea ice concentration.

12. Adding the sea-ice extend limit would also benefit the map.

**Authors' response:**
We appreciate the reviewer's suggestion and agree that adding the sea-ice extent (plotted at 15% concentration), would help to understand the sampling conditions better, particularly for emphasizing that the experiments presented in this manuscript were conducted in the sea-ice edge of the Dronning Maud Land region in autumn. Please see response to comment 11 above where we describe our decision to make a separate plot of the entire study region of the cruise.

13. « Underway measurements are shown with surface CTD data for (b-g) along with the initial incubation values » unclear

**Authors' response:**
This is indeed not clear. We have made the changes as per below and also clarify that these measurements were all conducted during March (please also see comment response #11):
**Original:**
"Underway measurements are shown with surface CTD data for (b-g) along with the initial incubation values."

**Modified:**
"The underway measurements, surface CTD data and the initial incubation measurements, all sampled within the study region in March, are collectively presented in (b-g)."

14. Table 1: Legend – "mean initial parameters for the photophysiology" mean of how many replicates? Please specify. "associated ancillary data" which are?

**Authors' response:**
**Original:**
"Sampling location information for the incubation stations with the associated CTD-Rosette water column station numbers from the cruise (CTD cast identifier) and mean initial parameters for the photophysiology (Fv/Fm and σPSII), as well as the associated ancillary data."

**Modified:**

"Sampling location information for the incubation stations with the associated CTD-Rosette water column station numbers from the cruise (CTD cast identifier) and mean (n=3) initial parameters for the photophysiology (Fv/Fm and σPSII), as well as the associated ancillary data (i.e., Chl-a concentrations, macronutrient concentrations and dFe concentrations)."

15. « Materials and methods » : remove Uppercase at Materials or harmonize

**Authors' response:**
Thank you, we have made the change as per below:
**Original:**
Table 1 caption:
"Cumulative photon dose and euphotic depth were calculated as defined in Materials and methods."

**Modified:**
"Cumulative photon dose and euphotic depth were calculated as defined in materials and methods."

16. "±" precedes standard deviation : how many replicates ?

**Authors' response:**
We have included n=3 to indicate the number of replicates.
**Original:**
Table 1 caption:
""n.d" indicates that no data was available and "±" precedes standard deviation."

**Modified:**
""n.d" indicates that no data was available and "±" precedes standard deviation (n=3)."

17. Cumulative photon dose : not clear – what depth ? what integration? within the MLD? Within 24h?

**Authors' response:**
We thank the reviewer for pointing this out and have subsequently corrected the "surface PAR measured" to read as "the PAR measured above the sea surface" and have clarified that it was summed over the duration of the experiment i.e. 24 hours.

**Original:**
"The cumulative photon dose for each experiment (mol photons $m^{-2}$ $d^{-1}$) was calculated as the cumulative sum of the surface PAR measured by a Biospherical Licor Chelsea PAR sensor on the ship's mast, starting from the time of experimental commencement, until experiment termination."

**Modified:**
"The cumulative photon dose for each experiment (mol photons m$^{-2}$ d$^{-1}$) was calculated as the cumulative sum of the PAR measured above the sea surface by a Biospherical Licor Chelsea PAR sensor on the ship's mast, starting from the time of experimental commencement, until experiment termination (i.e., summed over 24 hours)."

18. Sea Surface Temperature : I would suggest to write Sea Surface Temperature (SST) in legend and SST only in the Table. Or harmonize the spelling (uppercase)

**Authors' response:**
Thank you, we have made the changes as follows to the Table 1 caption, and corrected it as per the suggestion in the table for SST:
**Original:**
"Sea surface temperatures were obtained from the CTD sensor and were averaged for depths 15 to 30 m."

**Modified:**
"Sea Surface Temperatures (SSTs) were obtained from the CTD sensor and were averaged for depths 15 to 30 m."

19. Community Structure – if only dominant species are cited please specify. Structure or phytoplankton composition ?

**Authors' response:**
Table 1:
Thank you for the correction. The text has been updated to include "dominant phytoplankton community composition" in the caption, as well as in the table row heading "Dominant phytoplankton community composition".

**Original:**
"Community structure was taken from a combination of microscopy and CHEMTAX data from Kauko et al. (2022a; 2022b)"

**Modified:**
"Dominant phytoplankton community composition was taken from a combination of microscopy and CHEMTAX data from Kauko et al. (2022a; 2022b)."

**Materials and Methods**

20. Line 143: in situ in italic

**Authors' response:**
**Original:**
"In addition, initial in situ conditions for the incubation experiments from CTD surface samples are detailed below in section 2.3 (Incubation set-up and sub-sampling)."

**Modified:**
"In addition, initial _in situ_ conditions for the incubation experiments from CTD surface samples are detailed below in section 2.3 (Incubation set-up and sub-sampling)."

21. Line 147: 20-30 m depth is not surface anymore.

**Authors' response:**
When conducting trace metal sampling, a depth of 20 - 30 m is the shallowest one can safely sample 'trace metal clean water' with a GoFlo as 'surface seawater'.

Please see the following quote from Cutter and Bruland (2012):
"Moreover, the results for zinc and iron, in particular, show that the samples are uncontaminated below 20 m (sampling at shallower depths appears to result in contamination, perhaps from the ship's bottom paint and sacrificial zinc anodes)."

**References:**
Cutter, G.A., and Bruland, K.W.: Rapid and noncontaminating sampling system for trace elements in global ocean surveys. Limnology and Oceanography: Methods, 10(6), 425-436. DOI 10.4319/lom.2012.10.425, 2012.
https://aslopubs.onlinelibrary.wiley.com/doi/pdfdirect/10.4319/lom.2012.10.425

Thus, the only viable option currently available for shallower surface trace metal clean water (~2-5 m) is to sample using a trace clean FISH (GEOFISH). A trace metal clean FISH was used for sampling along the Weddell Sea during this cruise. However, it was most unfortunate that we could not guarantee a trace metal clean sampling and ran the risk of samples being contaminated.

22. Line 155: "bottles were filled unscreened" why not ? it should be at such depth to reproduce light attenuation.

**Authors' response:**

The screening in this context refers to the removal of grazers by filtering the seawater through a mesh. This has now been clarified as follows:

**Original:**
"These seven 1 L polycarbonate bottles were filled unscreened to represent 1 x the initial sample (hereafter 'initial'), 3 x the unamended control samples (hereafter 'Control'), and 3 x iron addition samples (hereafter 'Fe'), which were spiked with 2.0 nM iron (III) chloride (FeCl3 TraceCERT®; Sigma Aldrich) prepared in 2‰ HCl (30% suprapur HCl; Merck)."

**Modified:**
"These seven 1 L polycarbonate bottles were filled unscreened (i.e., no large grazers were excluded from the bottles) to represent 1 x the initial sample (hereafter 'initial'), 3 x the unamended control samples (hereafter 'Control'), and 3 x iron addition samples (hereafter 'Fe'), which were spiked with 2.0 nM iron (III) chloride (FeCl3 TraceCERT®; Sigma Aldrich) prepared in 2‰ HCl (30% suprapur HCl; Merck)."

23. Line 158: "spiked with 2.0 nM iron (III) chloride (FeCl3 TraceCERT®; Sigma Aldrich) prepared in 2‰ HCl (30% suprapur HCl; Merck)" – can the authors discuss the amount of Fe they think remained in solution (as dFe and therefore expected to be available to phytoplankton) after mixing with seawater?

**Authors' response:**
i) We note that Fe(III) precipitates at seawater pH. However, it may remain in the solution phase as complexes with dissolved organics (ligands) in seawater (Lannuzel et al., 2015). The quantification of this fraction, unfortunately, is not possible without knowing the ligand concentration (Smith et al., 2022), which is beyond the scope of this study.

**References:**
Lannuzel, D., Grotti, M., Abelmoschi, M.L., and Van Der Merwe, P.: Organic ligands control the concentrations of dissolved iron in Antarctic sea ice. Mar. Chem. 174, 120-130. https://doi.org/10.1016/j.marchem.2015.05.005, 2015. https://www.sciencedirect.com/science/article/pii/S0304420315001097

Smith, A.J., Nelson, T., Ratnarajah, L., Genovese, C., Westwood, K., Holmes, T.M., Corkill, M., Townsend, A.T., Bell, E., Wuttig, K., and Lannuzel, D.: Identifying potential sources of iron-binding ligands in coastal Antarctic environments and the wider Southern Ocean. Frontiers in Marine Science, 9, 948772. https://doi.org/10.3389/fmars.2022.948772, 2022. https://www.frontiersin.org/articles/10.3389/fmars.2022.948772/full

ii) The incubated trace metal clean seawater in this study was not iron-limited. Hence, not knowing the actual free iron added does not have any impact on the outcome of this study.

iii) Examples of previous literature that also used $FeCl_3$ for iron-addition incubation experiments without any reference to the fraction that remained available:

**References:**
Ryan-Keogh, T.J., Macey, A.I., Nielsdóttir, M.C., Lucas, M.I., Steigenberger, S.S., Stinchcombe, M.C., Achterberg, E.P., Bibby, T.S., and Moore, C.M.: Spatial and temporal development of phytoplankton iron stress in relation to bloom dynamics in the high-latitude North Atlantic Ocean. Limnol. Oceanogr. 58, 533–545. https://doi.org/10.4319/lo.2013.58.2.0533, 2013.

Li, Q., Legendre, L., and Jiao, N.: Phytoplankton responses to nitrogen and iron limitation in the tropical and subtropical Pacific Ocean. Journal of Plankton Research. 37(2), 306-319. https://doi.org/10.1093/plankt/fbv008, 2015.

We have altered the text as below to reduce ambiguity:

**Original:**
"These seven 1 L polycarbonate bottles were filled unscreened to represent 1 x the initial sample (hereafter 'initial'), 3 x the unamended control samples (hereafter 'Control'), and 3 x iron addition samples (hereafter 'Fe'), which were spiked with 2.0 nM iron (III) chloride ($FeCl_3$ TraceCERT®; Sigma Aldrich) prepared in 2‰ HCl (30% suprapur HCl; Merck)."

**Modified:**
"These seven 1 L polycarbonate bottles were filled unscreened to represent 1 x the initial sample (hereafter 'initial'), 3 x the unamended control samples (hereafter 'Control'), and 3 x iron addition samples (hereafter 'Fe'), which were spiked with iron (III) chloride ($FeCl_3$ TraceCERT®; Sigma Aldrich) prepared in 2‰ HCl (30% suprapur HCl; Merck), to reach a final concentration of 2.0 nM Fe."

24. Line 171: I am surprised there was only 43% of PAR with no screen.

**Authors' response:**
Indeed, we had run a few tests using different coverings (screens) over our incubators. This included first using 'blue lagoon' filters where the measured PAR was between 35 – 45 $\mu E \cdot m^{-2} \cdot s^{-1}$. This was followed by replacing the filter with a green mesh filter where the measured PAR remained ~45 $\mu E \cdot m^{-2} \cdot s^{-1}$. Eventually, we opted for no screen, as the PAR reaching the incubation bottles was limited by the autumn weather. The average PAR without any filter/screen was 230 $\mu E \cdot m^{-2} \cdot s^{-1}$ inside the polycarbonate bottles, which corresponded to 43% light passing through to the sample. Our understanding is that the bottle walls and the volume of water inside the bottle were able to reduce surface PAR by 43%.

25. Line 172: remove "experiment was terminated"

**Authors' response:**
**Original:**
"After each 24 hr period, the experiment was terminated and the incubation bottles removed from the incubator and sub-sampled under the clean, laminar flow hood (AirClean-600 PCR Workstation), inside the makeshift HEPA air-filtered Class-100 trace metal clean plastic bubble on-board as described above in section 2.2."

**Modified:**
"After each 24 hr period, the incubation bottles were removed from the incubator and sub-sampled under the clean, laminar flow hood (AirClean-600 PCR Workstation), inside the makeshift HEPA air-filtered Class-100 trace metal clean plastic bubble on-board as described above in section 2.2."

26. Line 178: what the authors consider relevant versus ancillary data?

**Authors' response:**
While both the "ancillary data" and the "relevant information" include additional data parameters that help better understand the initial conditions of the sampling locations for interpretation of the results in this manuscript, the "ancillary data" complements the photophysiological parameters (i.e., Fv/Fm and σPSII) for the initial and post-incubation results.

**Original:**

"A complete list of sampling locations, initial parameters for the photophysiology and ancillary data, as well as other relevant information (cumulative photon dose, MLD, euphotic depth and sea surface temperatures) is provided in Table 1."

We have now clarified the "ancillary data" parameters (repeated below) in the first paragraph of the "Materials and methods" section, as per the reviewer's suggestion (please see comment 10 from Line 115 in the preprint).

**Modified line 115:**

"Ancillary data (i.e., Chl-a concentrations, macronutrient concentrations and dFe concentrations) from surface water samples provide information on the regional conditions surrounding the five incubation experiments at the time of the cruise."

Furthermore, the "relevant information" in Line 178 is already mentioned in parenthesis in the text, i.e., " (cumulative photon dose, MLD, euphotic depth and sea surface temperatures)".

27. 2.4 misleading title as FRRf is not the acronym for Phytoplankton photosynthetic photophysiology. Remove FRRf and include Fast Repetition Rate Fluorometry in the text below.

**Authors' response:**
**Original:**
"2.4. Phytoplankton photosynthetic photophysiology (FRRf)"

**Modified:**
"2.4. Phytoplankton photosynthetic photophysiology"

Please note that the acronym "FRRf" has already been expanded in "2.3. Incubation set-up and sub-sampling", line 176:
"All incubation bottles were sub-sampled for photophysiological parameters using active Chl-a fluorescence measured through Fast Repetition Rate fluorometry (FRRf) (see section 2.4), Chl-a concentration (see section 2.5) and macronutrients (see section 2.6)."

28. 2.5 rename title as "Chlorophyll a (Chl a)"

**Authors' response:**
**Original:** "2.5 Chl-a"

**Modified:** "2.5 Chlorophyll-a (Chl-a)"

29. Line 204: "250 µL of chloroform » percentage ? at saturation ?

**Authors' response:**
**Original:**
**"**The seawater samples for macronutrient analysis (nitrate, phosphate and silicate) were collected in 50 mL Falcon tubes and preserved with 250 µL of chloroform."

**Modified:**
"The seawater samples for macronutrient analysis (nitrate, phosphate and silicate) were collected in 50 mL Falcon tubes for the incubation experiments and underway samples, whereas water column samples from the CTD-Rosette were collected in 20 ml vials. All samples were preserved with 250 µL of chloroform (saturated solution with 1% ethanol for stabilization). The samples were kept cold and in the dark until post-cruise analysis at the Institute of Marine Research, Bergen, Norway, using a colourimetric method (Grasshoff et al., 2009; Gundersen et al., 2022) on a Skalar autoanalyzer. The analyzer was calibrated using reference seawater from Ocean Scientific International Ltd. The detection limits were 0.5 µM for nitrate, 0.06 µM phosphate and 0.7 µM for silicate."

30. Line 204 : « the samples were kept cold » cold is a subjective word. In the fridge at 4°C ?

**Authors' response:**

**Original (modified from above):**
"The samples were kept cold and in the dark until post-cruise analysis at the Institute of Marine Research, Bergen, Norway, using a colourimetric method (Grasshoff et al., 2009; Gundersen et al., 2022) on a Skalar autoanalyzer."

**Modified:**
"The samples were kept cold (at 4°C in a fridge) and in the dark until post-cruise analysis at the Institute of Marine Research, Bergen, Norway, using a colourimetric method (Grasshoff et al., 2009; Gundersen et al., 2022) on a Skalar autoanalyzer."

**Results**

31. Line 260: If the authors are not citing their own results in Chla concentration, I suppose they cite average value for the MLD from Kauko et al. 2021? Please specify.

**Authors' response:**
The sentence does appear to be misleading. We have moved the reference earlier and have edited the Chl-a concentration part to indicate that it was indeed the measured value at the specific station location (Exp01).

**Original:**
"Despite being occupied in post-bloom conditions from a seasonal perspective (Fig. 2), Exp01 was, nonetheless, considered to represent autumn bloom conditions (albeit in decline) with high Chl-a concentrations (0.73 µg L$^{-1}$) (Kauko et al., 2021)."

**Modified:**
"Despite being occupied in post-bloom conditions from a seasonal perspective (Fig. 2), Exp01 was, nonetheless, considered to represent autumn bloom conditions (Kauko et al., 2021), albeit in decline, with a high Chl-a concentration (0.73 µg L$^{-1}$, Table 1)."

32. Line 267: I found excessive by the authors to say they "characterize two regions" with only one to two station in each of them but more importantly only one sample depth.

**Authors' response:**

Thank you for pointing this out as misleading, we clarify below that although there were limited numbers of experimental stations (n = 5) there were many more ancillary stations (n = 34). In addition, we have removed the word "characterize" and instead replaced it with "describe"

In line 267, we now state the following:
"Here, we first describe the general conditions in these two regions (n=34) and then focus specifically on the five experimental stations."

We emphasize that despite only having 2 experimental stations in the Astrid Ridge region and 3 experimental stations in the SIZ, we report on several parameters (i.e., Chl-a, Fv/Fm, $\sigma$PSII, nitrate, phosphate, silicate and dFe) for a much larger range of surface stations in each region (i.e., Astrid Ridge n=16 and SIZ n=18, please refer to Figure 1 b-h). In the same paragraph, we report the different surface ranges and means for the various parameters in the two regions and now have also included the results from comparing the above parameters using t-tests (please see comment response 33).
In lines 288 onward, we specifically go into the details of the initial conditions for the incubation stations.
Furthermore, these two subregions are further distinguished by patterns in the phenology, topographic and hydrographic features, which would affect the phytoplankton and the possibility of a bloom, noted in the companion publication by Kauko et al. (2021).

33. Line 270: I don't see differences in Fv/Fm between the two regions considering SD.

**Authors' response:**
**Original:**
"The mean values of Fv/Fm (Fig. 1c) were higher at Astrid Ridge (0.28±0.04) compared to the 6°E SIZ (0.24±0.06). The 6°E SIZ showed a much larger range in Fv/Fm with a minimum of 0.07 and a maximum of 0.34, whilst a narrower range in Fv/Fm, with a higher minimum in particular, was seen around Astrid Ridge (0.21 to 0.36)."

We had a run t-test on the Fv/Fm data between the surface Astrid Ridge and SIZ samples to assess if they were statistically significant. This was done using a Levene test to check for equal variances, A standard student's t-test was used if the data was of equal variance (which was the case for the Fv/Fm data between the two regions), while a Welch's t-test was

used for when data was of unequal variances.
The result here showed that the Fv/Fm between the Astrid Ridge and the SIZ were indeed statistically significant (p-value<0.05). We have thus included the 'p<0.05' in the text to clarify if the comparison was significantly different.

**Modified:**
"The mean values of Fv/Fm (Fig. 1c) were higher (p-value<0.05) at Astrid Ridge (0.28±0.04) compared to the 6°E SIZ (0.24±0.06). The 6°E SIZ showed a much larger range in Fv/Fm with a minimum of 0.07 and a maximum of 0.34, whilst a narrower range in Fv/Fm, with a higher minimum in particular, was seen around Astrid Ridge (0.21 to 0.36)."

34. Line 280: same remark for silicate concentrations…they are similar.

**Authors' response:**
Thank you for raising these similarities. We have compared the two regions for each of the macronutrients and dFe and have adjusted the manuscript text accordingly.

**Original:**
"Silicate concentrations showed a higher mean (48±1 µM) and less variability around Astrid Ridge with concentrations ranging from 46 to 52 µM, compared to a lower mean (46±2 µM) and larger range (41 to 49 µM) observed in the 6°E SIZ (Fig. 1g)."

Similarly, to the response above to comment #33, we applied a Levene t-test, and subsequently a student's t-test as the data was of equal variance for the surface Astrid Ridge and SIZ samples. The result was p-value<0.05, and hence the silicate concentrations in the two regions were significantly different.

**Modified:**
"Silicate concentrations showed a higher mean (48±1 µM, p-value<0.05) and less variability around Astrid Ridge with concentrations ranging from 46 to 52 µM, compared to a lower mean (46±2 µM) and larger range (41 to 49 µM) observed in the 6°E SIZ (Fig. 1g)."

To clarify for the other macronutrients:
Nitrate concentrations and dFe between the Astrid Ridge and SIZ were not significantly different (p-value>0.05), however, phosphate concentrations were significantly different (p-value<0.05). The text has thus been edited accordingly:

"Surface nitrate concentrations showed some spatial variability, but the mean values were similar (p-value>0.05) for the 6°E SIZ (mean 23.8±0.8 µM) and Astrid Ridge (mean 24.0±1.2 µM) (Fig. 1e). Despite a similarity in the range of phosphate concentrations observed for both the regions which ranged from 1.57 to 1.96 µM in the 6°E SIZ (mean 1.75±0.10 µM), and from 1.68 to 1.92 µM at Astrid Ridge (mean 1.82±0.06 µM) the regions were significantly different (p-value<0.05) (Fig. 1f). Silicate concentrations showed a higher mean (48±1 µM, p-value<0.05) and less variability around Astrid Ridge with concentrations ranging from 46 to 52 µM, compared to a lower mean (46±2 µM) and larger range (41 to 49 µM) observed in the 6°E SIZ (Fig. 1g). Despite the limited number of dFe measurements, a wide range of surface concentrations (Fig. 1h) were evident around Astrid Ridge with concentrations as low as 0.27 nM and as high as 1.39 nM (mean 0.64±0.49 nM). Mean dFe concentrations in the 6°E

SIZ were slightly lower (0.59±0.05 nM) compared to Astrid Ridge and varied over a narrow range between 0.56 to 0.63 nM. However, it is noted that these differences were not significant (p>0.05) and that only a fraction of the dFe is bioavailable to the phytoplankton, where this fraction can vary regionally and thus influence the variability in iron stress which may not mirror the ambient concentrations (Lis et al., 2015)."

35. Line 290: not the same location, therefore it is not a time-series. Using the word "drop" is therefore confusing.

**Authors' response:**
**Original:**
"Initial conditions in surface Chl-a ranged from high concentrations at the bloom station Exp01 (0.73 µg L$^{-1}$) dropping to as low as 0.02 µg L$^{-1}$ at Exp03 in the Astrid Ridge."

**Modified:**
"Initial conditions in surface Chl-a ranged from high concentrations at the bloom station Exp01 (0.73 µg L$^{-1}$), to concentrations as low as 0.02 µg L$^{-1}$ at Exp03 in the Astrid Ridge."

36. Line 311: I don't find Fv/Fm of 0.2 "particularly low". Lower than the others only.

**Authors' response:**

**Original:**
"The Fv/Fm was lower in the 6°E SIZ (mean 0.27±0.01) compared to Astrid Ridge (mean 0.35±0.01) and was particularly low at the bloom station Exp01 (0.20±0.01)."

**Modified:**
"The Fv/Fm was lower in the 6°E SIZ (mean 0.27±0.01) compared to Astrid Ridge (mean 0.35±0.01) and much lower at the bloom station Exp01 (0.20±0.01)."

37. Line 320: Change cholorophyll for Chl-a

**Authors' response:**

**Original:**
"Similarly, no significant differences (p>0.05) were observed in either macronutrient or chlorophyll concentrations (Table 2) between the Fe and Control incubations."

**Modified:**
"Similarly, no significant differences (p>0.05) were observed in either macronutrient or Chl–a concentrations (Table 2) between the Fe and Control incubations."

38. Figure 3: I would recommend to add on the top right of each panel the location of the stations to ease the reading. I would also add the ratio Fe/control to effectively compare the experiments together.

**Authors' response:**

As suggested, we have added the experimental number (i.e., Exp01, Exp02, etc.) at the top right of each panel in Figure 3. We will also incorporate the Fv/Fm ratio and σPSII of Fe:Control for each experiment in the corresponding panel.

39. Table 2: ratio between Fe/control would be helpful here as well. I recommend to add the consumption (delta between initial conditions and end of incubation) in macronutrients for each experiment.

**Authors' response:**

We appreciate the reviewer's recommendation to add the nutrient drawdown (consumption) between the initial and Control and between the initial and Fe for each experiment. We have indeed calculated this previously and present below the summary.
However, we note that there were no significant differences (p-value > 0.05) in the Δ(Nutrient) between the incubated Control and Fe samples for all three macronutrients. Thus, the nutrient drawdown, in this case, does not provide any additional information. As such, we would prefer not to include this additional information in the manuscript but provide it below for the reviewer.

**Calculating nutrient drawdown:**

Δ(Nutrient) = (Initial_Nutrient - Treatment_Nutrient)/Time (Units are: $\mu M\ d^{-1}$)

Time = 1 as the experiments where run for 24 hours

We obtained 3 Δ(Nutrient) for each of the control and the Fe samples. We then created a mean + stdev for both control and Fe which are presented below.

**Table. ΔNutrient Drawdown**

| | ΔNitrate [μM.d$^{-1}$]= (initial-control)/1 | ΔNitrate [μM.d$^{-1}$]= (initial-Fe)/1 | ΔPhosphate [μM.d$^{-1}$]= (initial-control)/1 | ΔPhosphate [μM.d$^{-1}$]= (initial-Fe)/1 | ΔSilicate [μM.d$^{-1}$]= (initial-control)/1 | ΔSilicate [μM.d$^{-1}$]= (initial-Fe)/1 |
|---|---|---|---|---|---|---|
| **Exp01** | -0.40±0.17 | -0.47±0.10 | 0.04±0.02 | -0.01±0.04 | 0.37±0.26 | 1.23±1.11 |
| **Exp02** | -0.17±0.33 | -0.07±0.53 | -0.02±0.01 | -0.01±0.01 | -1.57±1.00 | -0.97±0.05 |
| **Exp03** | -0.67±0.30 | -0.40±0.30 | -0.01±0.01 | 0.01±0.01 | -1.77±0.13 | -2.30±0.09 |
| **Exp04** | -0.20±0.17 | -1.57±0.48 | 0.05±0.01 | 0.02±0.01 | 0.53±0.05 | 0.97±0.48 |
| **Exp05** | -0.43±0.10 | -0.23±0.51 | -0.01±0.00 | -0.03x10$^{-1}$±0.01 | 1.33±0.05 | 1.07±0.20 |

**Discussion**

40. First paragraph (line 354-368): I would suggest to delete it as it sounds like a repetition of the introduction.

**Authors' response:**
We acknowledge that the first paragraph of the discussion section could rather be incorporated into the introduction section where relevant. We have adjusted accordingly by deleting the paragraph from the discussion.

41. Line 372: I am surprised by this statement. A large body of works on Fe uptake experiments in the Southern Ocean are conducted within 24h (ex. work by Strzepek and/or Fourquez). If the authors wish to keep this sentence please cite references to support this statement.

**Authors' response:**
We apologize for the misunderstanding stemming from our statement. The point of the statement was to highlight that the majority of incubation studies that have been conducted in the past, particularly including iron addition, were not dedicated solely to short-term incubations (24hrs), but rather, continued for longer timescales.
We do not intend to discredit long-term incubations, but rather suggest that based on the literature till date, short-term incubations can assist with rapid assessments of the phytoplankton photophysiology. At the same time, we did not intend to omit any existing publications on short-term incubations.

Furthermore, the work by Strzepek et al. (2011) is based predominantly on culture experiments, whilst the work of Fourquez et al. (2015) is focused on radioisotope uptake, which needs to be performed under very short time frames. Thus, we have clarified our statement with the modification as per below, as we do not consider iron uptake experiments in our manuscript.

**Original:**
"Furthermore, the majority of incubation studies were conducted as longer-term incubations (>96 hrs)."

"The majority of Southern Ocean incubation studies have shown that phytoplankton are iron-limited (de Baar et al., 1990; Viljoen et al., 2018; Ryan-Keogh et al., 2017a; 2018; Browning et al., 2014a; 2014b). However, no studies, to our knowledge, have been conducted in the SIZ during autumn. Furthermore, the majority of these iron addition incubation studies were conducted as longer-term incubations (>96 hrs). The complexity induced by longer-term nutrient addition incubations are exacerbated by artefacts that cause an isolated system to be devoid of natural factors such as, nutrient resupply and grazing which differs between the initial and incubated samples, whilst retaining only a specific sampled section from the water column as representative of the entire system (Geider and La Roche, 1994)."

---

## Author Response (AR2)

[revised manuscript text omitted]
., Watson, A., Bakker, D.C.E., Schuster, U., Metzl, N., Yoshikawa-Inoue, H., Ishii, M., Midorikawa, T., Nojiri, Y., Körtzinger, A., Steinhoff, T., Hoppema, M., Olafsson, J., Arnarson, T.S., Tilbrook, B., Johannessen, T., Olsen, A., Bellerby, R., Wong, C.S., Delille, B., Bates, N.R., and de Baar, H.J.W.D.Poisson, A.,
840 Metzl, N., Tilbrook, B., Bates, N., Wanninkhof, R., Feely, R.A., Sabine, C., Olafsson, J., and Yukihiro, N.: Global sea–air CO2 flux based on climatological surface ocean pCO2, and seasonal biological and temperature effects. Deep Sea Res. Part II Top. Stud. Oceanogr. 49, 1601–1622. https://doi.org/10.1016/S0967-0645(02)00003-6, 2002.

Takahashi, T., Sutherland, S.C., Wanninkhof, R., Sweeney, C., Feely, R.A., Chipman, D.W., Hales, B., Friederich, G., Chavez, F., Sabine, C., and Watson, A.: Climatological mean and decadal change in surface ocean pCO2, and net
845 sea–air CO2 flux over the global oceans. Deep Sea Res. Part II Top. Stud. Oceanogr. 56, 554–577. https://doi.org/10.1016/j.dsr2.2008.12.009, 2009.

Taylor, M.H., Losch, M., Bracher, A.: On the drivers of phytoplankton blooms in the Antarctic marginal ice zone: A modeling approach. J. Geophys. Res. Ocean. 118, 63–75. https://doi.org/10.1029/2012JC008418, 2013.

Thomalla, S.J., Fauchereau, N., Swart, S., and Monteiro, P.M.S.: Regional scale characteristics of the seasonal cycle of chlorophyll in the Southern Ocean. Biogeosciences, 8(10), 2849-2866. https://doi.org/10.5194/bg-8-2849-2011, 2011.

Trimborn, S., Thoms, S., Bischof, K., and Beszteri, S.: Susceptibility of two Southern Ocean phytoplankton key species to iron limitation and high light. Frontiers in Marine Science, 6, 167. https://doi.org/10.3389/fmars.2019.00167, 2019.

Van Oijen, T., Van Leeuwe, M.A., Granum, E., Weissing, F.J., Bellerby, R.G.J., Gieskes, W.W.C., and de Baar, H.J.W.: Light rather than iron controls photosynthate production and allocation in Southern Ocean phytoplankton populations during austral autumn. J. Plankton Res. 26, 885–900. https://doi.org/10.1093/plankt/fbh088, 2004.

Viljoen, J.J., Philibert, R., Van Horsten, N., Mtshali, T., Roychoudhury, A.N., Thomalla, S., and Fietz, S.: Phytoplankton response in growth, photophysiology and community structure to iron and light in the Polar Frontal Zone and Antarctic waters. Deep Sea Res. Part I Oceanogr. Res. Pap. 141, 118–129. https://doi.org/10.2495/EEIA100071, 2018.

Vink, S., and Measures, C.I.: The role of dust deposition in determining surface water distributions of Al and Fe in the South West Atlantic. Deep Sea Res. Part II Top. Stud. Oceanogr. 48, 2787–2809. https://doi.org/10.1016/S0967-0645(01)00018-2, 2001.

Wu, M., McCain, J.S.P., Rowland, E., Middag, R., Sandgren, M., Allen, A.E., and Bertrand, E.M.: Manganese and iron deficiency in Southern Ocean Phaeocystis antarctica populations revealed through taxon-specific protein indicators. Nature Communications, 10(1), 3582. https://doi.org/10.1038/s41467-019-11426-z, 2019.

Yoon, J.E., Yoo, K.C., Macdonald, A.M., Yoon, H.I., Park, K.T., Yang, E.J., Kim, H.C., Lee, J.I., Lee, M.K., Jung, J., and Park, J.: Reviews and syntheses: Ocean iron fertilization experiments–past, present, and future looking to a future Korean Iron Fertilization Experiment in the Southern Ocean (KIFES) project. Biogeosciences, 15(19), 5847-5889. https://doi.org/10.5194/bg-15-5847-2018, 2018.